# GSR: A Generalized Symbolic Regression Approach

**Tony Tohme**                                                        *tohme@mit.edu*
*Massachusetts Institute of Technology*

**Dehong Liu**                                                        *liudh@merl.com*
*Mitsubishi Electric Research Laboratories*

**Kamal Youcef-Toumi**                                               *youcef@mit.edu*
*Massachusetts Institute of Technology*

**Reviewed on OpenReview:** *https://openreview.net/forum?id=lheUXtDNvP*

## Abstract

Identifying the mathematical relationships that best describe a dataset remains a very challenging problem in machine learning, and is known as Symbolic Regression (SR). In contrast to neural networks which are often treated as black boxes, SR attempts to gain insight into the underlying relationships between the independent variables and the target variable of a given dataset by assembling analytical functions. In this paper, we present GSR, a Generalized Symbolic Regression approach, by modifying the conventional SR optimization problem formulation, while keeping the main SR objective intact. In GSR, we infer mathematical relationships between the independent variables and some transformation of the target variable. We constrain our search space to a weighted sum of basis functions, and propose a genetic programming approach with a matrix-based encoding scheme. We show that our GSR method is competitive with strong SR benchmark methods, achieving promising experimental performance on the well-known SR benchmark problem sets. Finally, we highlight the strengths of GSR by introducing SymSet, a new SR benchmark set which is more challenging relative to the existing benchmarks.

## 1 Introduction

Symbolic regression (SR) aims to find a mathematical expression that best describes the relationship between the independent variables and the target (or dependent) variable based on a given dataset. By inspecting the resulting expression, we may be able to identify nontrivial relations and/or physical laws which can provide more insight into the system represented by the given dataset. SR has gained tremendous interest and attention from researchers over the years for many reasons. First, many rules and laws in natural sciences (e.g. in physical and dynamical systems (Schmidt & Lipson, 2009; Quade et al., 2016)) are accurately represented by simple analytical equations (which can be explicit (Brunton et al., 2016) or implicit (Mangan et al., 2016; Kaheman et al., 2020)). Second, in contrast to neural networks that involve complex input-output mapping, and hence are often treated as black boxes which are difficult to interpret, SR is very concise and interpretable. Finally, symbolic equations may outperform neural networks in out-of-distribution generalization (especially for physical problems) (Cranmer et al., 2020).

SR does not require *a priori* specification of a model. Conventional regression methods such as least squares (Wild & Seber, 1989), likelihood-based (Edwards, 1984; Pawitan, 2001; Tohme et al., 2021), and Bayesian regression techniques (Lee, 1997; Leonard & Hsu, 2001; Tohme, 2020; Tohme et al., 2020; Vanslette et al., 2020) use fixed-form parametric models and optimize for the model parameters only. SR seeks to find both a model structure and its associated parameters simultaneously.

**Related Work.** The SR problem has been widely studied in the literature (Orzechowski et al., 2018; La Cava et al., 2021). SR can be a very challenging problem and is thought to be NP-hard (Lu et al., 2016; Udrescu & Tegmark, 2020; Petersen et al., 2021; Virgolin & Pissis, 2022). It can also be computationally expensive as the search space is very wide (or complex) containing expressions of any size and length de França (2018), this issue being exacerbated with the dimension of the input feature vector (i.e. the number of independent variables). Several approaches have been suggested over the years. Most of the methods use genetic (or evolutionary) algorithms (Koza & Koza, 1992; Schmidt & Lipson, 2009; Bäck et al., 2018; Virgolin et al., 2019). Some more recent methods are Bayesian in nature (Jin et al., 2019), some are physics-inspired (Udrescu & Tegmark, 2020), and others use divide-and-conquer (Luo et al., 2017) and block building algorithms (Chen et al., 2017b; 2018a;b). Lately, researchers proposed using machine learning algorithms and neural networks to solve the SR problem (Martius & Lampert, 2016; Sahoo et al., 2018; Udrescu et al., 2020; Ahn et al., 2020; Al-Roomi & El-Hawary, 2020; Kim et al., 2020; Kommenda et al., 2020; Burlacu et al., 2020; Biggio et al., 2021; Mundhenk et al., 2021; Petersen et al., 2021; Valipour et al., 2021; Razavi & Gamazon, 2022; Zhang et al., 2022a; d'Ascoli et al., 2022; Kamienny et al., 2022; Zhang et al., 2022b). Furthermore, some works suggested constraining the search space of functions to generalized linear space (Nelder & Wedderburn, 1972) (e.g. Fast Function eXtraction (McConaghy, 2011), Elite Bases Regression (Chen et al., 2017a), etc.) which proved to accelerate the convergence of genetic algorithms significantly (at the expense of sometimes losing the generality of the solution (Luo et al., 2017)).

Most of the SR methods use a tree-based implementation, where analytical functions are represented (or encoded) by expression trees. Some approaches suggested encoding functions as an integer string (O'Neill & Ryan, 2001), others proposed representing them using matrices (Luo & Zhang, 2012; Chen et al., 2017a; de França, 2018; de Franca & Aldeia, 2020). As we will discuss in later sections, our implementation relies on matrices to encode expressions.

**Our Contribution.** We present *Generalized Symbolic Regression* (GSR), by modifying the conventional SR optimization problem formulation, while keeping the main SR objective intact. In GSR, we identify mathematical relationships between the independent variables (or features) and some transformation of the target variable. In other words, we learn the mapping from the feature space to a transformed target space (where the transformation applied to the target variable is also learned during this process). To find the appropriate functions (or transformations) to be applied to the features as well as to the targets, we constrain our search space to a weighted sum of basis functions. In contrast to conventional tree-based genetic programming approaches, we propose a matrix-based encoding scheme to represent the basis functions (and hence the full mathematical expressions). We run a series of numerical experiments on the well-known SR benchmark datasets and show that our proposed method is competitive with many strong SR methods. Finally, we introduce SymSet, a new SR benchmark problem set that is more challenging than existing benchmarks.

## 2 Notation and Problem Formulation

Consider the following regression task. We are given a dataset $\mathcal{D} = \{\mathbf{x}_i, y_i\}_{i=1}^N$ consisting of $N$ i.i.d. paired examples, where $\mathbf{x}_i \in \mathbb{R}^d$ denotes the $i^{\text{th}}$ $d$-dimensional input feature vector and $y_i \in \mathbb{R}$ represents the corresponding continuous target variable. The goal of SR is to search the space of all possible mathematical expressions $\mathcal{S}$ defined by a set of given mathematical functions (e.g., exp, ln, sin, cos) and arithmetic operations (e.g., $+, -, \times, \div$), along with the following optimization problem:

$$f^* = \arg\min_{f \in \mathcal{S}} \sum_{i=1}^N \left[ f(\mathbf{x}_i) - y_i \right]^2 \tag{1}$$

where $f$ is the model function and $f^*$ is the optimal model.

# 3    Generalized Symbolic Regression (GSR)

In this section, we introduce our Generalized Symbolic Regression (GSR) approach. We present its problem formulation, and discuss its solution and implementation.

## 3.1    Modifying the goal of symbolic regression

As highlighted in Section 2, the goal of SR is to search the function space to find the model that best fits the mapping between the independent variables and the target variable (i.e. the mapping between $\mathbf{x}_i$ and $y_i$, for all $i$). Since the main objective of SR is to recognize correlations and find non-trivial interpretable models (rather than making direct predictions), we modify the goal of SR; we instead search the function space to find the model that best describes the mapping between the independent variables and a transformation of the target variable (i.e. the mapping between $\mathbf{x}_i$ and some transformation or function of $y_i$, for all $i$). Formally, we propose modifying the goal of SR to search for appropriate (model) functions from a space of all possible mathematical expressions $\mathcal{S}$ defined by a set of given mathematical functions (e.g., exp, ln, sin, cos) and arithmetic operations (e.g., $+, -, \times, \div$), which can be described by the following optimization problem:

$$f^*, g^* = \arg \min_{f, g \in \mathcal{S}} \sum_{i=1}^{N} \big[ f(\mathbf{x}_i) - g(y_i) \big]^2 \tag{2}$$

where $f^*$ and $g^*$ are the optimal analytical functions. In other words, instead of searching for mathematical expressions of the form $y = f(\mathbf{x})$ as is usually done in the SR literature, the proposed GSR approach attempts to find expressions of the form $g(y) = f(\mathbf{x})$. We illustrate this concept in Table 1.

Table 1: GSR finds analytical expressions of the form $g(y) = f(\mathbf{x})$ instead of $y = f(\mathbf{x})$.

| Ground Truth Expression | Learned Expression |
|---|---|
| $y = \sqrt{x + 5}$ | $y^2 = x + 5$ |
| $y = 1/(3x_1 + x_2^3)$ | $y^{-1} = 3x_1 + x_2^3$ |
| $y = (2x_1 + x_2)^{-\frac{2}{3}}$ | $\ln(y) = -\frac{2}{3} \ln(2x_1 + x_2)$ |
| $y = \ln(x_1^3 + 4x_1 x_2)$ | $e^y = x_1^3 + 4x_1 x_2$ |
| $y = e^{x_1^3 + 2x_2 + \cos(x_3)}$ | $\ln(y) = x_1^3 + 2x_2 + \cos(x_3)$ |

Although the main goal of GSR is to find expressions of the form $g(y) = f(\mathbf{x})$, we may encounter situations where it is best to simply learn expressions of the form $y = f(\mathbf{x})$ (i.e. $g(y) = y$). For instance, consider the ground truth expression $y = \sin(x_1) + 2x_2$. In this case, we expect to learn the expression exactly as is (i.e. $g(y) = y$ and $f(\mathbf{x}) = \sin(x_1) + 2x_2$) as long as the right basis functions (i.e. $\sin(x)$ and $x$ in this case) are within the search space, as we will see in the next sections.

**Making predictions.** Given a new input feature vector $\mathbf{x}_*$, predicting $y_*$ with GSR is simply a matter of solving the equation $g(y) = f(\mathbf{x}_*)$ for $y$, or equivalently, $g(y) - f(\mathbf{x}_*) = 0$. Note that $f(\mathbf{x}_*)$ is a known quantity and $y$ is the only unknown. If $g(\cdot)$ is an invertible function, then $y_*$ can be easily found using $y_* = g^{-1}\big(f(\mathbf{x}_*)\big)$. If $g(\cdot)$ is not invertible, then $y_*$ will be the root of the function $h(y) = g(y) - f(\mathbf{x}_*)$. Root-finding algorithms include Newton's method. Whether the function $g(\cdot)$ is invertible or not, we might end up with many solutions for $y_*$ (an invertible function, which is not one-to-one, can lead to more than one solution). In this case, we choose $y_*$ to be the solution that belongs to the range of $y$ which can be determined from the training dataset.

### 3.2 A new problem formulation for symbolic regression

Now that we have presented the goal of our proposed GSR approach (summarized by Equation 2), we need to constrain the search space of functions $\mathcal{S}$ to reduce the computational challenges and accelerate the convergence of our algorithm. Inspired by McConaghy (2011); Chen et al. (2017a) as well as classical system identification methods (Brunton et al., 2016), we confine $\mathcal{S}$ to generalized linear models, i.e. to functions that can be expressed as a linear combination (or as a weighted sum) of basis functions (which can be linear or nonlinear). In mathematical terms, for a given input feature vector $\mathbf{x}_i$ and a corresponding target variable $y_i$, the search space $\mathcal{S}$ is constrained to model functions of the form:

$$f(\mathbf{x}_i) = \sum_{j=1}^{M_\phi} \alpha_j \phi_j(\mathbf{x}_i), \qquad g(y_i) = \sum_{j=1}^{M_\psi} \beta_j \psi_j(y_i) \tag{3}$$

where $\phi_j(\cdot)$ and $\psi_j(\cdot)$ are the basis functions applied to the feature vector $\mathbf{x}_i$ and the target variable $y_i$, respectively, $M_\phi$ and $M_\psi$ denote the corresponding number of basis functions involved, respectively. In matrix form, the minimization problem described in Equation 2 is equivalent to finding the vectors of coefficients $\boldsymbol{\alpha} = [\alpha_1 \cdots \alpha_{M_\phi}]^T$ and $\boldsymbol{\beta} = [\beta_1 \cdots \beta_{M_\psi}]^T$ such that:

$$\boldsymbol{\alpha}^*, \boldsymbol{\beta}^* = \arg\min_{\boldsymbol{\alpha}, \boldsymbol{\beta}} ||\mathbf{X}\boldsymbol{\alpha} - \mathbf{Y}\boldsymbol{\beta}||^2 \tag{4}$$

where

$$\mathbf{X} = \begin{bmatrix} \phi_1(\mathbf{x}_1) & \phi_2(\mathbf{x}_1) & \cdots & \phi_{M_\phi}(\mathbf{x}_1) \\ \vdots & \vdots & \cdots & \vdots \\ \phi_1(\mathbf{x}_N) & \phi_2(\mathbf{x}_N) & \cdots & \phi_{M_\phi}(\mathbf{x}_N) \end{bmatrix}, \quad \mathbf{Y} = \begin{bmatrix} \psi_1(y_1) & \psi_2(y_1) & \cdots & \psi_{M_\psi}(y_1) \\ \vdots & \vdots & \cdots & \vdots \\ \psi_1(y_N) & \psi_2(y_N) & \cdots & \psi_{M_\psi}(y_N) \end{bmatrix}. \tag{5}$$

Note that if we examine the minimization problem as expressed in Equation 4, we can indeed minimize $||\mathbf{X}\boldsymbol{\alpha} - \mathbf{Y}\boldsymbol{\beta}||^2$ by simply setting $\boldsymbol{\alpha}^* = 0$ and $\boldsymbol{\beta}^* = 0$ which will not lead to a meaningful solution to our GSR problem. In addition, to avoid reaching overly complex mathematical expressions for $f(\cdot)$ and $g(\cdot)$, we are interested in finding sparse solutions for the weight vectors $\boldsymbol{\alpha}^*$ and $\boldsymbol{\beta}^*$ consisting mainly of zeros which results in simple analytical functions containing only the surviving basis functions (i.e. whose corresponding weights are nonzero). This is closely related to sparse identification of nonlinear dynamics (SINDy) methods (Brunton et al., 2016). To this end, we apply $L_1$ regularization, also known as *Lasso regression* (Tibshirani, 1996), by adding a penalty on the $L_1$ norm of the weights vector (i.e. the sum of its absolute values) which leads to sparse solutions with few nonzero coefficients. In terms of our GSR method, Lasso regression automatically performs basis functions selection from the set of basis functions that are under consideration.

Putting the pieces together, we reformulate the minimization problem in Equation 4 as a constrained Lasso regression optimization problem defined as

$$\boldsymbol{w}^* = \arg\min_{\boldsymbol{w}} ||\boldsymbol{A}\boldsymbol{w}||_2^2 + \lambda ||\boldsymbol{w}||_1$$
$$\text{s.t. } ||\boldsymbol{w}||_2 = 1 \tag{6}$$

where $\lambda > 0$ is the regularization parameter, and

$$\boldsymbol{A} = \begin{bmatrix} \boldsymbol{X} & -\boldsymbol{Y} \end{bmatrix}, \qquad \boldsymbol{w} = \begin{bmatrix} \boldsymbol{\alpha} \\ \boldsymbol{\beta} \end{bmatrix} = [\alpha_1 \cdots \alpha_{M_\phi} \ \beta_1 \cdots \beta_{M_\psi}]^T. \tag{7}$$

### 3.3 Solving the GSR problem

To solve the GSR problem, we first present our approach for solving the constrained Lasso problem in Equation 6, assuming some particular sets of basis functions are given. We then outline our genetic programming (GP) procedure for finding the appropriate (or optimal) sets of these basis functions, before discussing our matrix-based encoding scheme (to represent the basis functions) that we will use in our GP algorithm.

---

**Algorithm 1:** Solving the constrained Lasso optimization problem using ADMM

---

**Input:** $\boldsymbol{A}$, $\lambda$, $\rho$, $\boldsymbol{w}_0$, $\boldsymbol{z}_0$, $\boldsymbol{u}_0$
**Output:** $\boldsymbol{w}$
**function** SOLVEADMM($\boldsymbol{A}$, $\lambda$, $\rho$, $\boldsymbol{w}_0$, $\boldsymbol{z}_0$, $\boldsymbol{u}_0$)

    **Initialization:** $\boldsymbol{w} \leftarrow \boldsymbol{w}_0, \boldsymbol{z} \leftarrow \boldsymbol{z}_0, \boldsymbol{u} \leftarrow \boldsymbol{u}_0$;

    **while** *Not Converge* **do**

        $\boldsymbol{w} \leftarrow \left(2\boldsymbol{A}^T\boldsymbol{A} + \rho I\right)^{-1} \cdot \rho\left(\boldsymbol{z} - \boldsymbol{u}\right)$;

        $\boldsymbol{w} \leftarrow \boldsymbol{w}/||\boldsymbol{w}||_2$;

        $\boldsymbol{z} \leftarrow S_{\lambda/\rho}(\boldsymbol{w} + \boldsymbol{u})$;

        $\boldsymbol{u} \leftarrow \boldsymbol{u} + \boldsymbol{w} - \boldsymbol{z}$;

    **end**

**end function**

---

### 3.3.1 Solving the Lasso optimization problem given particular sets of basis functions

We assume for now that, in addition to the dataset $\mathcal{D} = \{\mathbf{x}_i, y_i\}_{i=1}^N$, we are also given the sets of basis functions $\{\phi_j(\mathbf{x}_i)\}_{j=1}^{M_\phi}$ and $\{\psi_j(y_i)\}_{j=1}^{M_\psi}$ used with the input feature vector $\mathbf{x}_i$ and its corresponding target variable $y_i$, respectively, for all $i$. In other words, we assume for now that the matrix $\boldsymbol{A}$ in Equation 7 is formed based on particular sets of basis functions (i.e. $\{\phi_j(\mathbf{x}_i)\}_{j=1}^{M_\phi}$ and $\{\psi_j(y_i)\}_{j=1}^{M_\psi}$), and we are mainly interested in solving the constrained optimization problem in Equation 6. Applying the alternating direction method of multipliers (ADMM) (Boyd et al., 2011), the optimization problem in Equation 6 can be written as

$$
\begin{aligned}
\boldsymbol{w}^* = \arg\min_{\boldsymbol{w}} \ &||\boldsymbol{A}\boldsymbol{w}||_2^2 + \lambda||\boldsymbol{z}||_1 \\
\text{s.t. } &||\boldsymbol{w}||_2 = 1 \\
&\boldsymbol{w} - \boldsymbol{z} = 0
\end{aligned}
\tag{8}
$$

where $\lambda > 0$ is the regularization parameter. The scaled form of ADMM (see Boyd et al. (2011) for details) for this problem is

$$
\begin{aligned}
\boldsymbol{w}_k &\leftarrow \arg\min_{||\boldsymbol{w}||_2=1} \mathcal{L}_\rho(\boldsymbol{w}, \boldsymbol{z}_{k-1}, \boldsymbol{u}_{k-1}) \\
\boldsymbol{z}_k &\leftarrow S_{\lambda/\rho}(\boldsymbol{w}_k + \boldsymbol{u}_{k-1}) \\
\boldsymbol{u}_k &\leftarrow \boldsymbol{u}_{k-1} + \boldsymbol{w}_k - \boldsymbol{z}_k
\end{aligned}
\tag{9}
$$

where $\boldsymbol{u}$ is the scaled dual vector, and

$$
\mathcal{L}_\rho(\boldsymbol{w}, \boldsymbol{z}, \boldsymbol{u}) = ||\boldsymbol{A}\boldsymbol{w}||_2^2 + \frac{\rho}{2}\left|\left|\boldsymbol{w} - \boldsymbol{z} + \boldsymbol{u}\right|\right|_2^2
\tag{10}
$$

where $\rho > 0$ is the penalty parameter and the *soft thresholding operator* $S$ is defined as

$$
S_\kappa(a) = \begin{cases} a - \kappa & a > \kappa \\ 0 & |a| \leq \kappa \\ a + \kappa & a < -\kappa \end{cases}
\tag{11}
$$

To find the minimizer $\boldsymbol{w}_k$ in the first step of the ADMM algorithm above (in Equation 9), we first compute the gradient of the function $\mathcal{L}_\rho(\boldsymbol{w}, \boldsymbol{z}_{k-1}, \boldsymbol{u}_{k-1})$ with respect to $\boldsymbol{w}$, set it to zero, and then normalize the resulting vector solution:

$$
\begin{aligned}
0 &= \nabla_{\boldsymbol{w}}\mathcal{L}_\rho(\boldsymbol{w}, \boldsymbol{z}_{k-1}, \boldsymbol{u}_{k-1})\big|_{\boldsymbol{w}=\boldsymbol{w}_k} \\
&= 2\boldsymbol{A}^T\boldsymbol{A}\boldsymbol{w}_k + \rho\left(\boldsymbol{w}_k - \boldsymbol{z}_{k-1} + \boldsymbol{u}_{k-1}\right)
\end{aligned}
\tag{12}
$$

It follows that

$$
\begin{aligned}
\boldsymbol{w}_k &= \left(2\boldsymbol{A}^T\boldsymbol{A} + \rho I\right)^{-1} \cdot \rho\left(\boldsymbol{z}_{k-1} - \boldsymbol{u}_{k-1}\right)\\
\boldsymbol{w}_k &= \boldsymbol{w}_k/\|\boldsymbol{w}_k\|_2
\end{aligned}
\tag{13}
$$

Algorithm 1 outlines the overall process for solving the constrained Lasso optimization problem in Equation 6, for a given matrix $\boldsymbol{A}$, regularization parameter $\lambda$, penalty parameter $\rho$, and initial guesses $\boldsymbol{w}_0$, $\boldsymbol{z}_0$, $\boldsymbol{u}_0$.

### 3.3.2 Finding the appropriate sets of basis functions using genetic programming

Now that we have presented Algorithm 1 that solves the constrained Lasso optimization problem in Equation 6 for particular sets of basis functions, we go through our procedure for finding the optimal sets of basis functions (and hence, the optimal analytical functions $f(\cdot)$ and $g(\cdot)$).

**Encoding Scheme**
Most of the SR methods rely on expression trees in their implementation. That is, each mathematical expression is represented by a tree where nodes (including the root) encode arithmetic operations (e.g. $+$, $-$, $\times$, $\div$) or mathematical functions (e.g. cos, sin, exp, ln), and leaves contain the independent variables (i.e. $x_1, \ldots, x_d$) or constants. Inspired by Luo & Zhang (2012); Chen et al. (2017a), we use matrices instead of trees to represent the basis functions. However, we propose our own encoding scheme that we believe is general enough to handle/recover a wide range of expressions.

We introduce the basis matrices $\boldsymbol{B}^\phi$ and $\boldsymbol{B}^\psi$ to represent the basis functions $\phi(\cdot)$ and $\psi(\cdot)$ used with the feature vector $\mathbf{x}$ and the target variable $y$, respectively. The basis matrices $\boldsymbol{B}^\phi$ and $\boldsymbol{B}^\psi$ are of sizes $n_{\boldsymbol{B}^\phi} \times m_{\boldsymbol{B}^\phi}$ and $n_{\boldsymbol{B}^\psi} \times 1$ respectively (i.e. $\boldsymbol{B}^\psi$ is a column vector), and take the form

$$
\boldsymbol{B}^\phi = \begin{bmatrix} b^\phi_{1,1} & \cdots & b^\phi_{1,m_{\boldsymbol{B}^\phi}} \\ \vdots & \cdots & \vdots \\ b^\phi_{n_{\boldsymbol{B}^\phi},1} & \cdots & b^\phi_{n_{\boldsymbol{B}^\phi},m_{\boldsymbol{B}^\phi}} \end{bmatrix}, \qquad \boldsymbol{B}^\psi = \begin{bmatrix} b^\psi_{1,1} \\ \vdots \\ b^\psi_{n_{\boldsymbol{B}^\psi},1} \end{bmatrix},
\tag{14}
$$

where the entries $b^\phi_{i,j}$ and $b^\psi_{i,1}$ are all integers. The first column $b^\phi_{\bullet,1}$ of $\boldsymbol{B}^\phi$ and the first (and only) column $b^\psi_{\bullet,1}$ of $\boldsymbol{B}^\psi$ indicate the mathematical function (or transformation) to be applied (on the input feature vector $\mathbf{x}$ and the target variable $y$, respectively). In $\boldsymbol{B}^\phi$, the second column $b^\phi_{\bullet,2}$ specifies the type of argument (see Table 2), and the remaining $n_v = m_{\boldsymbol{B}^\phi} - 2$ columns $b^\phi_{\bullet,3}, \cdots, b^\phi_{\bullet,m_{\boldsymbol{B}^\phi}}$ indicate which independent variables (or features) are involved (i.e. the active operands). The quantity $n_v$ represents the maximum total multiplicity of all the independent variables included in the argument. Note that $n_{\boldsymbol{B}^\phi}$ and $n_{\boldsymbol{B}^\psi}$ specify the number of transformations to be multiplied together (i.e. each basis function $\phi(\cdot)$ and $\psi(\cdot)$ will be a product of $n_{\boldsymbol{B}^\phi}$ and $n_{\boldsymbol{B}^\psi}$ transformations, respectively).

The encoding/decoding process happens according to a table of mapping rules that is very straightforward to understand and employ. For instance, consider the mapping rules outlined in Table 2, where $d$ is the dimension of the input feature vector. As we will see in our numerical experiments in Section 4, we will adopt this table for many SR benchmark problems. Other mapping tables are defined according to different benchmark problems[1] (more details about the SR benchmark problem specifications can be found in Appendix C). The encoding from the analytical form of a basis function to the basis matrix is straightforward. For example, for $d = 3$, $n_{\boldsymbol{B}^\phi} = 4$ and $m_{\boldsymbol{B}^\phi} = 5$ (i.e. $n_v = 3$), the basis function $\phi(\mathbf{x}) = x_2 \cos(x_1^2 x_2) \ln(x_1 + x_3)$ can be generated according to the encoding steps shown in Table 3.

---

[1]Each SR benchmark problem uses a specific set (or library) of allowable mathematical functions (e.g. cos, sin, exp, log), and hence, we mainly modify the first two rows of the mapping tables.

Table 2: Example table of mapping rules for a basis function. The identity operator is denoted by $\bullet^1$.

| $b_{\bullet,1}$ Transformation ($T$) | 0 | 1 | 2 | 3 | 4 | 5 |
|---|---|---|---|---|---|---|
| | 1 | $\bullet^1$ | cos | sin | exp | ln |

| $b_{\bullet,2}$ Argument Type ($arg$) | 0 | 1 | 2 |
|---|---|---|---|
| | $x$ | $\sum$ | $\prod$ |

| $b_{\bullet,3},\cdots,b_{\bullet,m_{\boldsymbol{B}}}$ Variable ($v$) | 0 | 1 | 2 | 3 | $\cdots$ | $d$ |
|---|---|---|---|---|---|---|
| | skip | $x_1$ | $x_2$ | $x_3$ | $\cdots$ | $x_d$ |

Table 3: Encoding steps corresponding to the basis function $\phi(\mathbf{x}) = x_2 \cos(x_1^2 x_2) \ln(x_1 + x_3)$.

| Step | $T$ | $arg$ | $v_1$ | $v_2$ | $v_3$ | Update |
|---|---|---|---|---|---|---|
| 1 | $\bullet^1$ | $x$ | $x_2$ | — | — | $T_1(\mathbf{x}) = x_2$ |
| 2 | cos | $\prod$ | $x_1$ | $x_1$ | $x_2$ | $T_2(\mathbf{x}) = \cos(x_1^2 x_2)$ |
| 3 | ln | $\sum$ | $x_1$ | $x_3$ | skip | $T_3(\mathbf{x}) = \ln(x_1 + x_3)$ |
| 4 | 1 | — | — | — | — | $T_4(\mathbf{x}) = 1$ |

Final Update: $\phi(\mathbf{x}) = T_1(\mathbf{x}) \cdot T_2(\mathbf{x}) \cdot T_3(\mathbf{x}) \cdot T_4(\mathbf{x})$

Based on the mapping rules in Table 2 and the encoding steps in Table 3, the basis function $\phi(\mathbf{x}) = x_2 \cos(x_1^2 x_2) \ln(x_1 + x_3)$ can be described by a $4 \times 5$ matrix as follows:

$$\boldsymbol{B}^\phi = \begin{bmatrix} 1 & 0 & 2 & \bullet & \bullet \\ 2 & 2 & 1 & 1 & 2 \\ 5 & 1 & 1 & 3 & 0 \\ 0 & \bullet & \bullet & \bullet & \bullet \end{bmatrix} \tag{15}$$

*Remark* 3.1. In Table 3, — denotes entries that are ignored during the construction of the basis function. The argument type in Step 1 is $x$ which implies that we only select the first variable (encoded by $b_{\bullet,3}$) out of the $n_v$ variables as an argument, and hence the entries corresponding to $v_2$ and $v_3$ are ignored. Similarly, the transformation in Step 4 is $T = 1$ which implies that the argument type and the $n_v$ variables are all ignored. These are the only two cases where some entries are ignored during the construction process. The same ignored entries are reflected in the matrix $\boldsymbol{B}^\phi$ using $\bullet$. More encoding examples can be found in Appendix B.

*Remark* 3.2. The term 'skip' can be thought of as 0 or 1 when the argument type is summation $\sum$ or multiplication $\prod$ respectively. To account for the case where the argument type is $x$, we let $b_{\bullet,3} \in \{1,\ldots,d\}$ (i.e. we exclude 0) as $b_{\bullet,3}$ is the only entry considered in this case (see *Remark* 3.1).

*Remark* 3.3. The same basis function can be represented by several matrices for three reasons:
i) Each basis function is a product of transformations where each transformation is represented by a row in the basis matrix. Hence, a new basis matrix for the same basis function is formed by simply swapping rows.
ii) When the argument type is $\sum$ or $\prod$, the order of the $n_v$ variables (including 'skip') starting from the third column of the matrix $\boldsymbol{B}^\phi$ does not affect the expression. Hence a new basis matrix for the same basis function is formed by simply swapping these columns.
iii) As mentioned in *Remark* 3.1, some entries are ignored in some cases. Hence a new basis matrix for the same basis function is formed by simply modifying these entries.

*Remark* 3.4. In the example above, we showed how we can produce the matrix $\boldsymbol{B}^\phi$ to represent a basis function $\phi(\mathbf{x})$. A similar (and even simpler) procedure can be applied to produce the matrix $\boldsymbol{B}^\psi$ that represents a basis function $\psi(y)$; we only need a mapping table corresponding to the set of allowable transformations (e.g. the first two rows of Table 2).

Note that the decoding from the basis matrix to the expression of a basis function is trivial; we go through the rows of the basis matrix and convert them into transformations according to a mapping table (e.g. Table 2), before finally multiplying them together. Also note that the search space of basis functions $\bigl(\text{mainly } \phi(\cdot)\bigr)$ is huge in general which makes enumeration impractical, and hence, we will rely on GP for effective search process.

**Genetic Programming (Evolutionary Algorithm)**

The SR problem has been extensively studied in the literature, and a wide variety of methods has been suggested over the years to tackle it. Most of these methods are based on genetic programming (GP) (Koza & Koza, 1992; Schmidt & Lipson, 2009; Bäck et al., 2018; Virgolin et al., 2019). This is a heuristic search technique that tries to find the optimal mathematical expression (in the SR context) among all possible expressions within the search space. The optimal (or best) expression is found by minimizing some objective function, known as the fitness function.

GP is an evolutionary algorithm that solves the SR problem. It starts with an initial population (or first generation) of $N_p$ randomly generated individuals (i.e. mathematical expressions), then recursively applies the *selection*, *reproduction (crossover)*, and *mutation* operations until *termination*. During the selection operation, the fitness of each of the $N_p$ individuals of the current generation is evaluated (according to the fitness function), and the $n_p$ fittest (or best) individuals are selected for reproduction and mutation (the selected individuals are part of the new generation and can be thought of as parents). The reproduction (crossover) operation generates new individuals (offsprings) by combining random parts of two parent individuals. The mutation operation produces a new individual by changing a random part of some parent individual. Finally, the recursion terminates, when some individual reaches a predefined fitness level (i.e. until some stopping criterion is satisfied).

In our GSR approach, we use a slightly modified version of the GP algorithm described above. Each individual in the population initially consists of two sets of $M_\phi$ and $M_\psi$ randomly generated basis functions encoded by basis matrices. Such matrices will form the functions $f(\cdot)$ and $g(\cdot)$ to be used with the input feature vector $\mathbf{x}$ and the target variable $y$, respectively. This is different from the GP algorithm described above where individuals typically represent the full mathematical expression or function as a whole. In addition, the $N_p$ individuals in the population of a new generation consist of the $n_p$ fittest individuals of the current generation in addition to $N_p - n_p$ individuals generated as follows. With probability $\frac{1}{4}$, a new individual is generated (reproduced) by randomly combining basis functions (i.e. basis matrices) from two parent individuals (i.e. crossover) selected from the $n_p$ surviving individuals. With probability $\frac{1}{4}$, a new individual is generated by randomly choosing one of the $n_p$ surviving individuals, and replacing (mutating) some of its basis functions (i.e. basis matrices) with completely new ones (i.e. randomly generated). With probability $\frac{1}{2}$, a completely new individual is randomly generated (in the same way we generate the individuals of the initial population). Randomly generating individuals enhances diversity in the basis functions and avoids reaching a plateau. Indeed, this is just one of many ways that can be followed to apply some sort of crossover/mutation on individuals defined by their sets of basis functions instead of their full mathematical expression. A pseudocode of our proposed GSR algorithm is provided in Appendix A.

## 4 Experimental Results

We evaluate our proposed GSR method through a series of numerical experiments on a number of common SR benchmark datasets. In particular, we compare our approach to existing state-of-the-art methods using three popular SR benchmark problem sets: Nguyen (Uy et al., 2011), Jin (Jin et al., 2019), and Neat (Trujillo et al., 2016). In addition, we demonstrate the benefits of our proposed method on the recently introduced SR benchmark dataset called Livermore (Mundhenk et al., 2021), which covers problems with a wider range of difficulty compared to the other benchmarks. Finally, we introduce a new and more challenging set of SR benchmark problems, which we call SymSet, mainly for two reasons: i) Our GSR algorithm achieves perfect scores on Nguyen, and almost perfect scores on Jin, Neat, and Livermore, and hence we introduced a benchmark problem set that is more challenging, ii) The existing SR benchmark problem sets do not really reflect the strengths of our proposed method, and thus we designed SymSet to explicitly highlight the benefits we gain from using our proposed approach. SymSet contains benchmark problems with similar properties as Nguyen, Jin, Neat, and Livermore benchmarks, but with an additional function composition (or symbolic layer). Each SR benchmark problem consists of a ground truth expression, a training and test dataset, and a set (or libary) of allowable arithmetic operations and mathematical functions. Specifications of all the SR benchmark problems are described in Appendix C. Hyperparameters and additional experiment details are provided in Appendix A.

Table 4: Recovery rate comparison of GSR against several algorithms on the Nguyen benchmark set over 100 independent runs. The formulas for these benchmarks are shown in Appendix Table 19.

| Benchmark | Expression | Recovery Rate (%) | | | |
|---|---|---|---|---|---|
| | | **GSR** | **NGGPPS** | **DSR** | **Eureqa** |
| Nguyen-1 | $y = x^3 + x^2 + x$ | 100 | 100 | 100 | 100 |
| Nguyen-2 | $y = x^4 + x^3 + x^2 + x$ | 100 | 100 | 100 | 100 |
| Nguyen-3 | $y = x^5 + x^4 + x^3 + x^2 + x$ | 100 | 100 | 100 | 95 |
| Nguyen-4 | $y = x^6 + x^5 + x^4 + x^3 + x^2 + x$ | 100 | 100 | 100 | 70 |
| Nguyen-5 | $y = \sin(x^2)\cos(x) - 1$ | 100 | 100 | 72 | 73 |
| Nguyen-6 | $y = \sin(x) + \sin(x + x^2)$ | 100 | 100 | 100 | 100 |
| Nguyen-7 | $y = \ln(x + 1) + \ln(x^2 + 1)$ | 100 | 97 | 35 | 85 |
| Nguyen-8 | $y = \sqrt{x}$ | 100 | 100 | 96 | 0 |
| Nguyen-9 | $y = \sin(x_1) + \sin(x_2^2)$ | 100 | 100 | 100 | 100 |
| Nguyen-10 | $y = 2\sin(x_1)\cos(x_2)$ | 100 | 100 | 100 | 64 |
| Nguyen-11 | $y = x_1^{x_2}$ | 100 | 100 | 100 | 100 |
| | **Average** | **100** | 99.73 | 91.18 | 80.64 |

Across our experiments, we compare our GSR approach against several strong SR benchmark methods:

**Neural-guided genetic programming population seeding (NGGPPS)**: A hybrid approach of neural-guided search and GP, which uses a recurrent neural network (RNN) to seed the starting population for GP (Mundhenk et al., 2021). NGGPPS achieves strong results on the well-known SR benchmarks.

**Deep Symbolic Regression (DSR)**: A reinforcement learning method that proposes a risk-seeking policy gradient to train an RNN to produce better-fitting expressions (Petersen et al., 2021). DSR is the "RNN only" version of NGGPPS, and is also considered a strong performer on the common SR benchmarks.

**Bayesian Symbolic Regression (BSR)**: A Bayesian framework which carefully designs prior distributions to incorporate domain knowledge (e.g. preference of basis functions or tree structure), and which employs efficient Markov Chain Monte Carlo (MCMC) methods to sample symbolic trees from the posterior distributions (Jin et al., 2019).

**Neat-GP**: a GP approach which uses the NeuroEvolution of Augmenting Topologies (NEAT) algorithm that greatly reduces the effects of bloat (i.e. controls the growth in program size) (Trujillo et al., 2016).

**PSTree**: A piece-wise non-linear SR method based on decision tree and GP techniques (Zhang et al., 2022a). PSTree can generate explainable models with high accuracy in a short period of time. PSTree is the current top performer on SRBench datasets (La Cava et al., 2021), achieving state-of-the-art performance and beating other competitive SR methods such as Operon (Kommenda et al., 2020; Burlacu et al., 2020) and AI Feynman (Udrescu et al., 2020).

**PySR**: A fast and parallelized SR method in Python/Julia (Cranmer, 2020), which uses evolutionary algorithms to search for symbolic expressions by optimizing a particular objective; the metric used for scoring equations is based on the work by Cranmer et al. (2020).

**gplearn**: A Koza-style SR method in Python, which starts with a random population of models, and then iteratively performs tournament selection, crossover, and mutation (Koza & Koza, 1992).

We first compare GSR against NGGPPS, DSR, as well as Eureqa (a popular GP-based commercial software proposed in Schmidt & Lipson (2009)) on the Nguyen benchmarks. We follow their experimental procedure and report the results in Table 4. We use *recovery rate* as our performance metric, defined as the fraction of independent training runs in which an algorithm's resulting expression achieves exact symbolic equivalence compared to the ground truth expression (as verified using a computer algebra system such as SymPy (Meurer et al., 2017)). Table 4 shows that GSR significantly outperforms DSR and Eureqa in exactly recovering the Nguyen benchmark expressions. As NGGPPS achieves nearly perfect scores on the Nguyen benchmarks, GSR shows only a slight improvement (on Nguyen-7) compared to NGGPPS. However, GSR exhibits faster runtime than NGGPPS; by running each benchmark problem, GSR takes an average of 2.5 minutes per run on the Nguyen benchmarks compared to 3.2 minutes for NGGPPS. Runtimes on individual Nguyen benchmark problems are shown in Appendix Table 11.

Table 5: Comparison of mean root-mean-square error (RMSE) for GSR against several methods on the Jin benchmark problem set over 50 independent runs. The formulas for these benchmarks are shown in Appendix Table 19.

| Benchmark | Mean RMSE | | | |
|-----------|------|--------|---------|------|
| | GSR | NGGPPS | DSR | BSR |
| Jin-1 | 0 | 0 | 0.46 | 2.04 |
| Jin-2 | 0 | 0 | 0 | 6.84 |
| Jin-3 | 0 | 0 | 0.00052 | 0.21 |
| Jin-4 | 0 | 0 | 0.00014 | 0.16 |
| Jin-5 | 0 | 0 | 0 | 0.66 |
| Jin-6 | 0.018 | 0 | 2.23 | 4.63 |
| **Average** | 0.0030 | **0** | 0.45 | 2.42 |

Table 6: Comparison of median RMSE for GSR against several methods on the Neat benchmark problem set over 30 independent runs. The formulas for these benchmarks are shown in Appendix Table 19.

| Benchmark | Median RMSE | | | |
|-----------|------|--------|---------|---------|
| | GSR | NGGPPS | DSR | Neat-GP |
| Neat-1 | 0 | 0 | 0 | 0.0779 |
| Neat-2 | 0 | 0 | 0 | 0.0576 |
| Neat-3 | 0 | 0 | 0.0041 | 0.0065 |
| Neat-4 | 0 | 0 | 0.0189 | 0.0253 |
| Neat-5 | 0 | 0 | 0 | 0.0023 |
| Neat-6 | $2.0 \times 10^{-4}$ | $6.1 \times 10^{-6}$ | 0.2378 | 0.2855 |
| Neat-7 | 0.0521 | 1.0028 | 1.0606 | 1.0541 |
| Neat-8 | $4.0 \times 10^{-4}$ | 0.0228 | 0.1076 | 0.1498 |
| Neat-9 | $8.1 \times 10^{-9}$ | 0 | 0.1511 | 0.1202 |
| **Average** | **0.0059** | 0.1139 | 0.1756 | 0.1977 |

We next evaluate GSR on the Jin and Neat benchmark sets. The results are reported in Tables 5 and 6 respectively. A RMSE value of 0 indicates exact symbolic equivalence. From Table 5, we can clearly observe that GSR outperforms DSR and BSR and performs nearly as good as NGGPPS recovering all the Jin problems (accross all independent runs) except Jin-6. Table 6 shows that GSR outperforms all other methods (NGGPPS, DSR, and Neat-GP) on the Neat benchmarks. Note that expressions containing divisions (i.e. Neat-6, Neat-8, and Neat-9) are not exactly recovered by GSR (i.e. only approximations are recovered) since the division operator is not included in our scheme (see Appendix E for details).

We then run experiments on the Livermore benchmark set which contains problems with a large range of difficulty. In addition to NGGPPS and DSR, we compare against NGGPPS using the soft length prior (SLP) and hierarchical entropy regularizer (HER) recently introduced in Larma et al. (2021). We also compare against a recently proposed method, known by genetic expert-guided learning (GEGL) (Ahn et al., 2020), which trains a molecule-generating deep neural network (DNN) guided with genetic exploration. Table 7 shows that our GSR method outperforms all other methods on both the Nguyen and Livermore benchmark sets, beating NGGPPS+SLP/HER which was the top performer on these two benchmark sets.

We highlight the strengths of GSR on the new SymSet benchmark problem set, and show the benefits of searching for expressions of the form $g(y) = f(\mathbf{x})$ instead of $y = f(\mathbf{x})$. Typical expressions, with exact symbolic equivalence, recovered by GSR are shown in Appendix Table 26. The key feature of GSR lies in its ability to recover expressions of the form $g(y) = f(\mathbf{x})$. To better highlight the benefits offered by this feature, we disable it by constraining the search space in GSR to expressions of the form $y = f(\mathbf{x})$ (which is the most critical ablation). We refer to this special version of GSR as s-GSR. Note that most of the SymSet expressions cannot be exactly recovered by s-GSR (i.e. they can only be approximated). We compare the performance of GSR against s-GSR on the SymSet benchmarks in terms of accuracy and runtime (see Table 8). The results clearly show that GSR is faster than s-GSR, averaging around 2 minutes per run on the SymSet benchmarks compared to 2.27 minutes for s-GSR (i.e. $\sim 11\%$ runtime improvement). In addition, GSR is more accurate than s-GSR by two orders of magnitude. This is due to the fact that GSR exactly recovers the SymSet expressions across most of the runs, while s-GSR only recovers approximations for most of these expressions. This reflects the superiority of GSR over s-GSR, which demonstrates the benefits of learning expressions of the form $g(y) = f(\mathbf{x})$ in SR tasks. We further compare GSR against several strong SR methods with similar (or better) expression ability. In particular, we experiment on SymSet with NGGPPS, PSTree, PySR, and gplearn (see Table 8). GSR is more accurate than all these methods by three orders of magnitude, which further demonstrates the advantage of our proposed approach. As for the runtime, PSTree is the fastest method, averaging around 16 seconds per run on the SymSet expressions, while maintaining solid accuracies. This comes as no surprise given its state-of-the-art performance on SRBench datasets (La Cava et al., 2021).

Table 7: Recovery rate comparison of GSR against several algorithms on the Nguyen and Livermore benchmark sets over 25 independent runs. Recovery rates on individual benchmark problems are shown in Appendix Table 10.

| | Recovery Rate $(\%)$ | | |
| --- | --- | --- | --- |
| | All | Nguyen | Livermore |
| **GSR** | **90.59** | **100.00** | **85.45** |
| **NGGPPS+SLP/HER** | 82.59 | 92.00 | 77.45 |
| **NGGPPS** | 78.59 | 92.33 | 71.09 |
| **GEGL** | 66.82 | 86.00 | 56.36 |
| **DSR** | 49.18 | 83.58 | 30.41 |

Table 8: Average performance in mean RMSE and runtime, along with their standard errors, for GSR against s-GSR and several strong SR methods on the SymSet benchmark problem sets over 25 independent runs. Mean RMSE and runtime values on individual benchmark problems are shown in Appendix Table 12.

| | **SymSet Average** | | |
| --- | --- | --- | --- |
| | Mean RMSE | | Runtime (sec) |
| **GSR** | $\mathbf{2.66 \times 10^{-4}}$ $\pm$ $\mathbf{1.59 \times 10^{-4}}$ | | 120.84 $\pm$ 4.22 |
| **s-GSR** | $2.56 \times 10^{-2}$ $\pm$ $5.27 \times 10^{-3}$ | | 136.19 $\pm$ 4.56 |
| **PSTree** | $4.57 \times 10^{-1}$ $\pm$ $7.98 \times 10^{-2}$ | | $\mathbf{16.23}$ $\pm$ $\mathbf{0.67}$ |
| **NGGPPS** | $4.65 \times 10^{-1}$ $\pm$ $1.24 \times 10^{-1}$ | | 158.57 $\pm$ 2.59 |
| **PySR** | $4.99 \times 10^{-1}$ $\pm$ $1.76 \times 10^{-1}$ | | 87.07 $\pm$ 21.6 |
| **gplearn** | $7.22 \times 10^{-1}$ $\pm$ $1.64 \times 10^{-1}$ | | 163.86 $\pm$ 2.94 |

## 5 Discussion

**Limitations.** GSR, including state-of-the-art methods, have difficulty with expressions containing divisions. For GSR, this is due to the way we define our encoding scheme. Other methods fail even though the division is included in their framework. GSR can overcome this issue by modifying its encoding scheme to include divisions within the basis functions (at the expense of significantly increasing the complexity of the search space). Another limiting factor to GSR is that it cannot recover expressions containing composition of functions, such as $y = e^{\cos(x)} + \ln(x)$. This could be overcome by modifying the search space (e.g. one could expand the definition of a basis function to account for composition of functions up to some number of layers, or completely modify the search space to a symbolic neural network as in Martius & Lampert (2016); Sahoo et al. (2018); Kim et al. (2020)). Another challenging task for GSR is to reach, although expressible, expressions containing multiple complex basis functions simultaneously. This can be due to the choice of the hyperparameters or the GP search process. A more elaborate discussion about the limitations of GSR can be found in Appendix E. These limitations will be addressed in a future paper. Indeed, there are plenty of expressions that still cannot be fully recovered by GSR. This is the case for all other SR methods as well.

**Closely related work.** There has been growing attention on the SR task with non explicit (or implicit) mathematical equations and several works have been attempted to address this interesting task. In particular, implicit sparse identification of nonlinear dynamics (implicit-SINDy) (Mangan et al., 2016; Kaheman et al., 2020) introduces the concept of identifying implicit expressions of the form $f(\mathbf{x}, y) = 0$ in the context of differential equations $\left(\text{i.e. } y = \dot{x}_i = \frac{dx_i}{dt} \text{ for } i \in \{1, \ldots, d\} \text{ where } \mathbf{x} \in \mathbb{R}^d\right)$. Further, Eureqa (Schmidt & Lipson, 2009), a well-established baseline SR algorithm, focuses on discovering invariants rather then trying to perform prediction directly. Inspired by the two aforementioned methods, and by the fact that the main objective of SR is to recognize correlations and define non-trivial interpretable models, GSR identifies relations between the input and a transformed output through searching for expressions of the form $g(y) = f(\mathbf{x})$, which keeps the possibility open for predicting the output $y$ in a straightforward manner.

**Computational complexity.** Although genetic algorithms are inherently heuristic, understanding how our GSR algorithm operates and scales could still be valuable. Following Algorithm 2 from Appendix A, we can approximate the time complexity of GSR as:

$$O\left(N_\epsilon \cdot N_p \cdot \left(M_\phi \cdot n_{\boldsymbol{B}^\phi} \cdot m_{\boldsymbol{B}^\phi} + M_\psi \cdot n_{\boldsymbol{B}^\psi} + N_\delta + N + \log N_p\right)\right) \tag{16}$$

where $N_\epsilon$ is the number of generations until the GP algorithm converges, and $N_\delta$ is the number of iterations until the ADMM algorithm converges. Recall that $N_p$ is the population size, $M_\phi$ and $M_\psi$ denote the number of $n_{\boldsymbol{B}^\phi} \times m_{\boldsymbol{B}^\phi}$ and $n_{\boldsymbol{B}^\psi} \times 1$ basis matrices applied to $\mathbf{x}$ and $y$, respectively, and $N$ is the number of paired training examples. More details about GSR's computational complexity can be found in Appendix A.

Compared to GSR, the special version s-GSR adopts a vanilla SR (where $g(y)$ is simply $y$) with the same GP algorithm and coefficient optimization process (through ADMM) as GSR. Hence, s-GSR's time complexity can be approximated as:

$$O\Big(N_\epsilon \cdot N_p \cdot \big(M_\phi \cdot n_{\boldsymbol{B}^\phi} \cdot m_{\boldsymbol{B}^\phi} + N_\delta + N + \log N_p\big)\Big) \tag{17}$$

Although GSR's computational complexity contains an additional term of $O\left(N_\epsilon \cdot N_p \cdot M_\psi \cdot n_{\boldsymbol{B}^\psi}\right)$, the number of GP generations $N_\epsilon$ produced by GSR is often much less than that of s-GSR, which explains the runtime advantage of GSR over s-GSR shown in Table 8.

**GSR's expression ability.** The term *Generalized* in GSR mainly stands for its ability to discover analytical mappings from the input space to a transformed output space through expressions of the form $g(y) = f(\mathbf{x})$. This generalizes the classical SR task of identifying expressions of the form $y = f(\mathbf{x})$ (i.e. the latter is simply a special case of GSR with $g(y) = y$). In addition, the term *Generalized* can denote the fact that we constrain the search space to generalized linear models, keeping in mind that the search space could be confined to other generalized spaces. Note that, by finding relations of the form $g(y) = f(\mathbf{x})$, the expression ability of GSR could resemble that of classical SR tasks which search for relations $y = f_c(x)$ where the composition function $f_c(\cdot)$ is defined as $f_c(\cdot) := h \circ f(\cdot) = h(f(\cdot))$ with $h(\cdot) = g^{-1}(\cdot)$ if $g(\cdot)$ is invertable, or a function class of similar expression ability as $g^{-1}(\cdot)$ if $g(\cdot)$ is not invertable. However, GSR takes advantage of the fact that the target $y$ is a scalar, and hence, we can apply many basis functions to $y$ (through $g(\cdot)$) without much increasing the complexity of the expression. In other words, we can avoid searching for functions equivalent to $g^{-1}(\cdot)$ in a space that could grow exponentially with the dimension of the input feature vector by simply searching for their corresponding inverse transformations applied to the scalar target variable. This concept, which happens implicitly in our algorithm provides an edge for GSR over traditional SR methods in terms of runtime, complexity, and smoothness of the search space. In short, GSR discovers simplified expressions by reducing redundancies in the search space, which greatly saves the computational complexity of the search process. It is worth mentioning that, in principle, GSR's concept of fitting $g(y) = f(\mathbf{x})$ (instead of $y = f(\mathbf{x})$) could be applied in conjunction with other classical SR methods; this may require some modifications to their parameter/coefficient optimization process.

**GSR: a simple yet promising algorithm.** GSR combines features and benefits from the usually disparate fields of system identification and genetic programming. On the one hand, SINDy methods use some LASSO-like approaches (or sequential thresholded least squares) to conduct their sparse non-linear regression for finding solutions that take the form of a linear combination of basis functions. On the other hand, evolutionary algorithms are effective in finding basis functions that achieve optimal solution. In other words, GSR combines well established evolutionary methods with more classical system identification methods. Although each of the algorithm components are relatively simple, the overall GSR algorithm achieves promising experimental performance, highlighting new insights, which can open up new research directions for future improvement.

## 6    Conclusion

We introduce GSR, a Generalized Symbolic Regression approach by modifying the formulation of the conventional SR optimization problem. In GSR, we identify mathematical relationships between the input features and some transformation of the target variable. That is, we infer the mapping from the feature space to a transformed target space, by searching for expressions of the form $g(y) = f(\mathbf{x})$ instead of $y = f(\mathbf{x})$. We confine our search space to a weighted sum of basis functions and use genetic programming with a matrix-based encoding scheme to extract their expressions. We perform several numerical experiments on well-known SR benchmark datasets and show that our GSR approach is competitive with strong SR benchmark methods. We further highlight the strengths of GSR by introducing SymSet, a new SR benchmark set which is more challenging relative to the existing benchmarks. In principle, GSR's concept of fitting $g(y) = f(\mathbf{x})$ could be extended to existing SR methods and could boost their performance.

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

# A  Implementation, Hyperparameters, and Additional Experiment Details

**Implementation.** Our GSR method discovers expressions of the form $g(y) = f(\mathbf{x})$ where $y \in \mathbb{R}$, $\mathbf{x} \in \mathbb{R}^d$, and where the search space for $f(\cdot)$ and $g(\cdot)$ is constrained to a weighted sum of $M_\phi$ and $M_\psi$ basis functions (namely $\phi(\cdot)$ and $\psi(\cdot)$), respectively. We use a matrix-based encoding scheme to represent $\phi(\cdot)$ and $\psi(\cdot)$ using basis matrices $\boldsymbol{B}^\phi$ and $\boldsymbol{B}^\psi$ of sizes $n_{\boldsymbol{B}^\phi} \times m_{\boldsymbol{B}^\phi}$ and $n_{\boldsymbol{B}^\psi} \times 1$, respectively (where $m_{\boldsymbol{B}^\phi} = n_v + 2$ and $n_v$ is defined in Section 3.3.2). Hence, in addition to the number of basis functions $M_\phi$ and $M_\psi$, the parameters $n_{\boldsymbol{B}^\phi}$, $n_v$, and $m_{\boldsymbol{B}^\phi}$ affect the complexity of the evolved expressions, and hence can be controlled to confine the search space (although $d$ also affects the complexity of the expression, it is given by the problem and cannot be controlled). Although more than one basis matrix can lead to the same basis function (see *Remark* 3.3), the search space of basis functions (mainly $\phi(\cdot)$) is still huge in general, and thus, enumerating all the possible basis functions is not practical. Hence, we will rely on genetic programming (GP) for effective search process. A pseudocode of our GP-based GSR algorithm is outlined in Algorithm 2.

---

**Algorithm 2:** GP Procedure for GSR

---

**Input:** $N_p$, $n_p$, $M_\phi$, $M_\psi$, $\mathcal{L}_\mathbf{x}$, $\mathcal{L}_y$
**Output:** $\mathcal{I}^*$
**function** SolveGSR($N_p$, $n_p$, $M_\phi$, $M_\psi$, $\mathcal{L}_\mathbf{x}$, $\mathcal{L}_y$)
    **Initialize population:**      // $\boldsymbol{\mathcal{I}}(k) \leftarrow \{\mathcal{I}_1(k), \mathcal{I}_2(k), \dots, \mathcal{I}_{N_p}(k)\}$
    $k \leftarrow 0$;      // Initialize the generation (or iteration) counter
    **for** $i = 1$ **to** $N_p$ **do**
        $\mathcal{I}_i(k) \leftarrow$ GenerateRandomIndividual($M_\phi$, $M_\psi$, $\mathcal{L}_\mathbf{x}$, $\mathcal{L}_y$);
        /* Each individual $\mathcal{I}_i(k)$ contains two randomly generated sets of $M_\phi$ and $M_\psi$ basis matrices respectively   */
    **end**
    Evaluate each individual $\mathcal{I}_i(k)$ with respect to the fitness function;
    /* For each individual $\mathcal{I}_i(k)$, form the matrix $A_i(k)$, solve for the optimal coefficients vector $\boldsymbol{w}_i(k) \leftarrow$ SolveADMM($A_i(k), \cdots$), then compute its fitness   */
    $\boldsymbol{\mathcal{I}}(k) \leftarrow$ sorted($\boldsymbol{\mathcal{I}}(k)$);      // in ascending order of fitness
    **while** *Stopping Criterion not Satisfied* **do**
        $k \leftarrow k + 1$;      // Increment the generation (or iteration) counter
        UpdateCriterion();
        /* Start with a strict stopping criterion (e.g. a very low error threshold) and slowly relax it (e.g. gradually increase the error threshold)   */
        $\mathcal{L}_\mathbf{x}^s, \mathcal{L}_y^s \leftarrow$ ChooseSublibrary($\mathcal{L}_\mathbf{x}, \mathcal{L}_y$);
        /* Choose sublibraries $\mathcal{L}_\mathbf{x}^s \subseteq \mathcal{L}_\mathbf{x}$ and $\mathcal{L}_y^s \subseteq \mathcal{L}_y$ of allowable operations to be used with $\mathbf{x}$ and $y$ respectively   */
        $\boldsymbol{\mathcal{I}}_{[1:n_p]}(k) \leftarrow \boldsymbol{\mathcal{I}}_{[1:n_p]}(k-1)$;
        /* The $n_p$ fittest individuals of the previous generation are copied to the current new one   */
        **for** $i = n_p + 1$ **to** $N_p$ **do**
            $u \leftarrow$ GenerateRandomInteger($1, 4$);
            **if** $u = 1$ **then**
                $\mathcal{I}_i(k) \leftarrow$ Reproduce($\boldsymbol{\mathcal{I}}_{[1:n_p]}(k), \mathcal{L}_\mathbf{x}^s, \mathcal{L}_y^s$);
                /* Crossover based on the surviving individuals   */
            **else if** $u = 2$ **then**
                $\mathcal{I}_i(k) \leftarrow$ Mutate($\boldsymbol{\mathcal{I}}_{[1:n_p]}(k), \mathcal{L}_\mathbf{x}^s, \mathcal{L}_y^s$);
                /* Mutation based on the surviving individuals   */
            **else**
                $\mathcal{I}_i(k) \leftarrow$ GenerateRandomIndividual($M_\phi$, $M_\psi$, $\mathcal{L}_\mathbf{x}^s, \mathcal{L}_y^s$);
                /* Randomly generate a completely new individual   */
            **end**
        **end**
        Evaluate each individual $\mathcal{I}_i(k)$ with respect to the fitness function;
        $\boldsymbol{\mathcal{I}}(k) \leftarrow$ sorted($\boldsymbol{\mathcal{I}}(k)$);      // in ascending order of fitness
    **end**
    $\mathcal{I}^* \leftarrow \mathcal{I}_1(k)$;      // return the fittest individual
**end function**

---

The main inputs to our GP-based algorithm are $N_p$, $n_p$, $M_\phi$, $M_\psi$, $\mathcal{L}_\mathbf{x}$, and $\mathcal{L}_y$. Recall that $N_p$ is the population size and $n_p$ is the number of surviving individuals per generation. $\mathcal{L}_\mathbf{x}$ and $\mathcal{L}_y$ are the libraries of allowable transformations that can be used with $\mathbf{x}$ and $y$, respectively. These libraries form the first

two rows of mapping tables, e.g. Table 13, and are defined by the benchmark problem. Note that the division operator is not part of our GSR architecture. That is, the main arithmetic operations used by GSR are $\{+, -, \times\}$. For example, for the Nguyen benchmark dataset, the library of allowable operations is $\mathcal{L}_0 = \{+, -, \times, \div, \cos, \sin, \exp, \ln\}$ as shown in Table 19. In this case, we define $\mathcal{L}_{\mathbf{x}} = \{1, \bullet^1, \cos, \sin, \exp, \ln\}$ (resulting in the mapping Table 2) and $\mathcal{L}_y = \{1, \bullet^1, \exp, \ln\}$. Regarding the stopping criterion, common terminating conditions for GP include: i) a solution reaches minimum criterion (e.g. error threshold), ii) the algorithm reaches a fixed number of generations (or iterations), iii) the algorithm generates a fixed number of individuals (or candidates expressions), iv) the algorithm reaches a plateau such that new generations no longer improve results, v) combinations of the above conditions. In our case, the algorithm terminates when the solution hits a minimum root-mean-square error (RMSE) threshold. To accelerate termination, we slowly relax the error threshold by gradually increasing it. To avoid reaching a plateau and since we are dealing with a small population size as shown in Table 9, we enhance diversity (in the basis functions) by producing completely new individuals with probability $1/2$ per generation (while performing crossover and mutation with probability $1/4$ each per generation). To speed up the search process, we employ sublibraries $\mathcal{L}_{\mathbf{x}}^s \subseteq \mathcal{L}_{\mathbf{x}}$ and $\mathcal{L}_y^s \subseteq \mathcal{L}_y$ of allowable transformations, used when generating completely new individuals (or completely new basis functions in the case of mutation). For $\mathbf{x}$, we mainly rely on three sublibraries which are the most common: a polynomial sublibrary $\mathcal{L}_{\text{poly}}$, a trigonometric sublibrary $\mathcal{L}_{\text{trig}}$, and the original library $\mathcal{L}_{\mathbf{x}}$ itself. For the Nguyen benchmark example above, $\mathcal{L}_{\text{poly}} = \{1, \bullet^1\}$ and $\mathcal{L}_{\text{trig}} = \{1, \bullet^1, \cos, \sin\}$. Note that power operators such as $\bullet^2$, $\bullet^3$ would be included in these sublibraries if they were part of the original library defined by the benchmark problem. The function CHOOSESUBLIBRARY( ) works according to some cycle. For example, assuming $k$ is the generation (or iteration) counter, if $k \leq 1,500$, each cycle consists of 70 iterations broken into three stages, the first stage consists of 15 iterations and assigns $\mathcal{L}_{\mathbf{x}}^s \leftarrow \mathcal{L}_{\mathbf{x}}$, the second stage consists of 25 iterations and assigns $\mathcal{L}_{\mathbf{x}}^s \leftarrow \mathcal{L}_{\text{poly}}$, and the third and final stage consists of the remaining 35 iterations and assigns $\mathcal{L}_{\mathbf{x}}^s \leftarrow \mathcal{L}_{\text{trig}}$. This cycle repeats until $k = 1,500$, after which the cycle's size becomes $1,500$ iterations broken into three equal stages (i.e. 500 iterations per sublibrary). For $y$, the cycle consists of 20 iterations, in which we equally alternate between the polynomial sublibrary $\mathcal{L}_{\text{poly}}$ and the original library $\mathcal{L}_y$ itself (i.e. 10 iterations for each sublibrary). Indeed, the use of sublibraries is only possible when the corresponding operations are included in the original library defined by the benchmark problem (e.g. Neat-6 and Neat-8 cannot use trigonometric sublibraries since $\{\cos, \sin\}$ are not included in their corresponding original libraries, as shown in Table 19). In addition, it is up to the user to specify the cycle's size and how to alternate between sublibraries, or even decide whether to use sublibraries in the first place.

**Hyperparameters.** Throughout our experiments, we adopt the following hyperparameter values. For GP, we use a population size $N_p = 30$, and we allow for $n_p = 10$ surviving individuals per generation. We perform crossover with probability $P_c = \frac{1}{4}$ and allow for only 2 parents to be involved in the process (i.e. new individuals are formed by combing basis functions from two randomly chosen parent individuals). We apply mutation with probability $P_m = \frac{1}{4}$ and allow for 3 basis functions (randomly selected from an individual) to be mutated (i.e. to be discarded and replaced by completely new basis functions). We generate a (completely new) random individual with probability $P_r = \frac{1}{2}$. For ADMM, we use a regularizer $\lambda = 0.4$, a penalty $\rho = 0.1$. The algorithm terminates when the $\ell^2$-norm of the difference between the weight vectors from two consecutive iterations falls below a threshold of $\delta = 10^{-5}$. Regarding initial conditions, we use $\boldsymbol{w}_0 = \widehat{\frac{1}{2}} = \frac{[\frac{1}{2} \cdots \frac{1}{2}]^T}{\sqrt{\frac{1}{4} + \cdots + \frac{1}{4}}}$ (where "$\widehat{\phantom{x}}$" denotes a normalized vector), $\boldsymbol{z}_0 = \mathbf{1} = [1 \cdots 1]^T$, $\boldsymbol{u}_0 = \mathbf{0} = [0 \cdots 0]^T$. For GSR, we allow for a maximum of $M_\phi = 15$ basis functions $\phi(\cdot)$ for each expression of $f(\cdot)$ (this is the maximum number since some of the $M_\phi$ basis functions will be multiplied by 0, i.e. at most we get $M_\phi$ nonzero coefficients multiplying the basis fcuntions). To avoid overfitting and overly complex expressions, we allow for a maximum of $M_\psi = 1$ basis function $\psi(\cdot)$ for each expression of $g(\cdot)$ (in this case the maximum and minimum are both 1 and $g(\cdot)$ will consist of a single basis function). It is worth noting that we use $M_\psi = 2$ for SymSet-11. Each basis $\psi(\cdot)$ will consist of a single transformation $n_{\boldsymbol{B}^\psi} = 1$. Each basis $\phi(\cdot)$ will be a product of $N_t$ transformations, where $N_t$ is a random integer between 1 and 3, i.e. $n_{\boldsymbol{B}^\phi} \in \{1, 2, 3\}$. For each of these $N_t$ transformations, the maximum total multiplicity of all the independent variables (or features) in an argument is a random integer between 2 and 5, i.e. $n_v \in \{2, 3, 4, 5\}$. GSR terminates when a candidate expression achieves a RMSE lower than a threshold with a starting value of $\epsilon = 10^{-6}$ (recall that this threshold is slowly relaxed during the process, e.g. by progressively multiplying it by a factor of $\sqrt{10}$ for every $1,500$ iterations). All hyperparameter values are summarized in Table 9.

Table 9: Hyperparameter values for GSR for all experiments, unless otherwise specified.

| Hyperparameter | Symbol | Value |
|---|---|---|
| **GP Parameters** | | |
| Population size | $N_p$ | 30 |
| Number of survivors per generation | $n_p$ | 10 |
| Crossover probability | $P_c$ | 1/4 |
| Number of parents involved in crossover | — | 2 |
| Mutation probability | $P_m$ | 1/4 |
| Number of bases to mutate | — | 3 |
| Randomly generated individual probability | $P_r$ | 1/2 |
| **ADMM Parameters** | | |
| Regularization parameter | $\lambda$ | 0.4 |
| Penalty parameter | $\rho$ | 0.1 |
| Tolerance on the solution error | $\delta$ | $10^{-5}$ |
| Initial guesses | $\boldsymbol{w}_0, \boldsymbol{z}_0, \boldsymbol{u}_0$ | $\widehat{\frac{1}{2}}, \mathbf{1}, \mathbf{0}$ |
| **GSR Parameters** | | |
| Maximum number of basis functions $\phi(\cdot)$ for each expression of $f(\cdot)$ | $M_\phi$ | 15 |
| Maximum number of basis functions $\psi(\cdot)$ for each expression of $g(\cdot)$ | $M_\psi$ | 1 |
| Maximum total multiplicity of all features in an argument | $n_v$ | $\{2, 3, 4, 5\}$ |
| Number of tranformations multiplied together per basis $\phi(\cdot)$ | $n_{\boldsymbol{B}^\phi}$ | $\{1, 2, 3\}$ |
| Number of tranformations multiplied together per basis $\psi(\cdot)$ | $n_{\boldsymbol{B}^\psi}$ | 1 |
| Tolerance on the solution error (RMSE) | $\epsilon$ | $10^{-6}$ |

**Computational complexity.** Although genetic algorithms are inherently heuristic, understanding how our GSR algorithm operates and scales could still be valuable. Following Algorithm 2 above, we can approximate the time complexity of GSR as:

$$
O\Bigg( N_p \cdot \Big( M_\phi \cdot n_{\boldsymbol{B}^\phi} \cdot m_{\boldsymbol{B}^\phi} + M_\psi \cdot n_{\boldsymbol{B}^\psi} \cdot 1 + N_\delta + O\left(\text{fitness}\right) \Big) + O\left(N_p \log N_p\right)
$$
$$
+ N_\epsilon \cdot \Big( (N_p - n_p) \cdot \big( P_c \cdot O\left(\text{crossover}\right) + P_m \cdot O\left(\text{mutation}\right) + P_r \cdot (M_\phi \cdot n_{\boldsymbol{B}^\phi} \cdot m_{\boldsymbol{B}^\phi} + M_\psi \cdot n_{\boldsymbol{B}^\psi} \cdot 1)
$$
$$
+ N_\delta + O\left(\text{fitness}\right) \big) + O\left(N_p \log N_p\right) \Big) \Bigg) \tag{18}
$$

where $N_\epsilon$ is the number of generations until the GP algorithm hits the tolerance $\epsilon$, and $N_\delta$ is the number of iterations until the ADMM algorithm hits the tolerance $\delta$. Note that performing crossover or mutation operations takes $O(1)$ time (i.e. a constant amount of time), and computing the fitness (which calculates RMSE on $N$ paired training examples) takes $O(N)$ time. Also note that $P_c$, $P_m$, and $P_r$ are probabilities which can be treated as constants. Hence, GSR's time complexity reduces to:

$$
O\Big( N_\epsilon \cdot N_p \cdot \big( M_\phi \cdot n_{\boldsymbol{B}^\phi} \cdot m_{\boldsymbol{B}^\phi} + M_\psi \cdot n_{\boldsymbol{B}^\psi} + N_\delta + N + \log N_p \big) \Big) \tag{19}
$$

Compared to GSR, the special version s-GSR adopts a vanilla SR (where $g(y)$ is simply $y$) with the same GP algorithm and coefficient optimization process (through ADMM) as GSR. Thus, s-GSR's time complexity can be approximated as:

$$
O\Big( N_\epsilon \cdot N_p \cdot \big( M_\phi \cdot n_{\boldsymbol{B}^\phi} \cdot m_{\boldsymbol{B}^\phi} + N_\delta + N + \log N_p \big) \Big) \tag{20}
$$

Although GSR's computational complexity contains an additional term of $O\left(N_\epsilon \cdot N_p \cdot M_\psi \cdot n_{\boldsymbol{B}^\psi}\right)$, the number of GP generations $N_\epsilon$ produced by GSR is often much less than that of s-GSR, which explains the runtime advantage of GSR over s-GSR shown in Table 12.

**Additional experiment details.** For all benchmark problems, we run GSR for multiple independent trials using different random seeds (following the experimental procedure in Petersen et al. (2021); Mundhenk et al. (2021)). Table 10 shows the recovery rates of GSR against literature-reported values from several algorithms on the Nguyen and Livermore individual benchmark problems. We first note that, due to the wide domain of sampled input points imposed by Livermore-1 (i.e. $[-10, 10]$), we observed some instabilities in the solution due to the presence of the exponential function, which we decided to exclude from the library of allowable operations during the search process for this benchmark. In what follows, we provide explanations for the results shown in Table 10. Note that Livermore-5 is difficult to recover by NGGPPS+SLP/HER and the remaining methods as it contains subtractions. Subtraction is more difficult than addition since it is not cumulative. This is not an issue for GSR since both additions and subtractions are equally recovered through the sign of the optimal coefficients multiplying the basis functions. Livermore-10 and Livermore-17 are more challenging than Nguyen-10 since they require adding the same basis function many more times (which is apparent through the poor recovery rates of the different methods). Fortunately, this is also not a problem for GSR since it can be easily solved by finding the right coefficient multiplying the basis function. Livermore-18 is more challenging than Livermore-2 and Nguyen-5 since it requires recovering the constant 5 without a constant optimizer (which can be recovered as $\frac{x+x+x+x+x}{x}$). For GSR, this can be recovered by naturally solving for the real-valued coefficient. The problem of the different methods on Livermore-22 lies in the constant 0.5, which requires finding $\frac{x}{x+x}$ compared to GSR which simply solves for the optimal parameter multiplying $x^2$. We observe that GSR performs poorly on Livermore-9 and Livermore-21 compared to NGGPPS+SLP/HER. This can be due to the choice of hyperparameters (e.g. $M_\phi$, $n_{\boldsymbol{B}^\phi}$, and $n_v$) as well as the GP-based search process. These two benchmarks require finding the first 9 and 8 powers of $x$ simultaneously, respectively, which can be difficult to achieve by GSR, especially that we only consider $M_\phi = 15$ basis functions $\phi(\cdot)$ per expression of $f(\cdot)$, as mentioned earlier. Note that polynomials were not an issue for GSR up to the $6^{\text{th}}$ order (i.e. Nguyen-4). We also tried experimenting with a $7^{\text{th}}$ order polynomial (i.e. $y = x^7 + x^6 + x^5 + x^4 + x^3 + x^2 + x$) and GSR achieved 100% recovery rate. We started observing a decline in the recovery rate when we added the $8^{\text{th}}$ power of $x$. In other words, Livermore-21 (the $8^{\text{th}}$ order polynomial) seems to be the limit that GSR can reach with polynomials while Livermore-9 (the $9^{\text{th}}$ order polynomial) becomes very difficult to recover. It is worth noting that if the libraries for the Livermore-9 and Livermore-21 problems contained the square and cube operators $\{\bullet^2, \bullet^3\}$ (as is the case for the Jin benchmarks described in Table 19), then GSR would easily recover these two problems. Finally, GSR is not able to recover Livermore-7 $\big(y = \sinh(x)\big)$ and Livermore-8 $\big(y = \cosh(x)\big)$ since both benchmarks require finding the basis function $e^{-\bullet}$,[2] which cannot be expressed using our current encoding scheme unless it is available as a transformation by itself. That is, the exponential operator $e^\bullet$ is not enough to recover $\frac{1}{e^\bullet} = e^{-\bullet}$ using our current encoding scheme. Had the negative exponential operator $e^{-\bullet}$ been part of the library of allowable operations defined by Livermore-7 and Livermore-8, GSR would easily recover these two benchmarks. As we can see for SymSet-1 $\big(y = x\sinh(x) - \frac{4}{5}\big)$, we added the operator $e^{-\bullet}$ to the library of allowable operations (see Table 20), which made GSR's mission much simpler and it was able to recover the corresponding ground truth expression as shown in Table 26. It is worth mentioning that, although ground truth expressions are not expressible, GSR was naturally able to recover the best approximations possible for Livermore-7 and Livermore-8, which turned out to be their Taylor expansions around 0. GSR's typical output expressions were as follows:

Livermore-7:

$$
\begin{aligned}
0.51655\,y &= +0.51655x + 0.48165x(x \times x) + 0.48165x(x \times x) \\
&\quad - 0.048738(x + x + x)(x + x + x)(x + x) + 0.0022335(x + x)(x)(x \times x \times x) \\
\iff \quad y &\approx x + 0.166x^3 + 0.00865x^5 \\
&\approx x + \frac{x^3}{3!} + \frac{x^5}{5!} \qquad \big(\text{the first three terms of the Taylor series of } \sinh(x) \text{ around } 0\big) \\
&\approx \sinh(x)
\end{aligned}
$$

---

[2]Livermore-7 and Livermore-8 can be expressed as $\sinh(x) = \frac{e^x - e^{-x}}{2}$ and $\cosh = \frac{e^x + e^{-x}}{2}$ respectively.

Livermore-8:

$$-0.8616\,y = -0.2154 - 0.042956(x+x)x - 0.035877(x \times x)(x \times x) - 0.2154 - 0.2154$$
$$- 0.0012368(x \times x)(x \times x \times x \times x) - 0.042956x(x+x) - 0.25898x(x) - 0.2154$$

$$\iff \quad y \approx 1 + 0.5x^2 + 0.0416x^4 + 0.00144x^6$$

$$\approx 1 + \frac{x^2}{2!} + \frac{x^4}{4!} + \frac{x^6}{6!} \qquad \left(\text{the first four terms of the Taylor series of } \cosh(x) \text{ around } 0\right)$$

$$\approx \cosh(x)$$

We next perform a runtime comparison between GSR and NGGPPS on the Nguyen benchmark problem set. We run each benchmark problem and report the runtimes in Table 11. We find that GSR exhibits faster runtime than NGGPPS, averaging 2.5 minutes per run on the Nguyen benchmarks compared to 3.2 minutes for NGGPPS. It is worth noting that although GSR is, on average, faster than NGGPPS, it still exhibits slower runtime on some problems (e.g. Nguyen-6 and Nguyen-9 in Table 11). This is due to the randomness of the search process as well as the use of sublibraries as mentioned earlier in the Appendix. Indeed, the runtime depends on the stopping criterion or condition. For example, one can shorten the runtime further if the interest is just in an approximation rather than an exact recovery. Our GSR method recovers exact expressions in the order of few minutes.

We further highlight the strengths of GSR on the new SymSet benchmark problem set, and show the benefits of searching for expressions of the form $g(y) = f(\mathbf{x})$ instead of $y = f(\mathbf{x})$. Typical expressions, with exact symbolic equivalence, recovered by GSR are shown in Table 26. The key feature of GSR lies in its ability to recover expressions of the form $g(y) = f(\mathbf{x})$. To better highlight the benefits offered by this feature, we disable it by constraining the search space in GSR to expressions of the form $y = f(\mathbf{x})$ (which is the most critical ablation). We refer to this special version of GSR as s-GSR. Note that most of the SymSet expressions cannot be exactly recovered by s-GSR (i.e. they can only be approximated). We compare the performance of GSR against s-GSR on the SymSet benchmarks in terms of accuracy and runtime (see Table 12). The results clearly show that GSR is faster than s-GSR, averaging around 2 minutes per run on the SymSet benchmarks compared to 2.27 minutes for s-GSR (i.e. $\sim 11\%$ runtime improvement). In addition, GSR is more accurate than s-GSR by two orders of magnitude. This is due to the fact that GSR exactly recovers the SymSet expressions across most of the runs, while s-GSR only recovers approximations for most of these expressions. It is worth mentioning that on SymSet-1, SymSet-4, SymSet-5, SymSet-10, and SymSet-12, we observe mean RMSE values of the same order of magnitude between GSR and s-GSR, since these expressions can be exactly recovered by simply learning expressions of the form $y = f(\mathbf{x})$. As GSR has to perform a search to discover that $g(y)$ is simply $y$ for these expressions, it exhibits slower runtime than s-GSR in recovering these expressions (see Table 12).

In addition, we compare GSR against several strong SR methods with similar (or better) expression ability. In particular, we experiment on SymSet with NGGPPS, PSTree, PySR, and gplearn (see Table 12). GSR is more accurate than all these methods by three orders of magnitude, which further demonstrates the advantage of our proposed approach. As for the runtime, PSTree is the fastest method, averaging around 16 seconds per run on the SymSet expressions, while maintaining solid accuracies. This comes as no surprise given its state-of-the-art performance on SRBench datasets (La Cava et al., 2021). It is worth mentioning that on SymSet-16, all the methods (i.e. NGGPPS, PSTree, PySR, and gplearn) exhibited some instabilities in the solution over all independent runs. Hence, we excluded SymSet-16 for these methods in Table 12.

Table 10: Recovery rate comparison of GSR against literature-reported values from several algorithms on the Nguyen and Livermore benchmark problem sets over 25 independent runs. The ground truth expressions for these benchmarks are shown in Tables 19 and 20.

| Benchmark | Recovery Rate (%) | | | | |
|---|---|---|---|---|---|
| | **GSR** | **NGGPPS + SLP/HER** | **NGGPPS** | **GEGL** | **DSR** |
| Nguyen-1 | 100 | 100 | 100 | 100 | 100 |
| Nguyen-2 | 100 | 100 | 100 | 100 | 100 |
| Nguyen-3 | 100 | 100 | 100 | 100 | 100 |
| Nguyen-4 | 100 | 100 | 100 | 100 | 100 |
| Nguyen-5 | 100 | 100 | 100 | 92 | 72 |
| Nguyen-6 | 100 | 100 | 100 | 100 | 100 |
| Nguyen-7 | 100 | 100 | 96 | 48 | 35 |
| Nguyen-8 | 100 | 100 | 100 | 100 | 96 |
| Nguyen-9 | 100 | 100 | 100 | 100 | 100 |
| Nguyen-10 | 100 | 100 | 100 | 92 | 100 |
| Nguyen-11 | 100 | 100 | 100 | 100 | 100 |
| Nguyen-12* | 100 | 4 | 12 | 0 | 0 |
| **Nguyen Average** | **100** | 92.00 | 92.33 | 86.00 | 83.58 |
| Livermore-1 | 100 | 100 | 100 | 100 | 3 |
| Livermore-2 | 100 | 100 | 100 | 44 | 87 |
| Livermore-3 | 100 | 100 | 100 | 100 | 66 |
| Livermore-4 | 100 | 100 | 100 | 100 | 76 |
| Livermore-5 | 100 | 40 | 4 | 0 | 0 |
| Livermore-6 | 100 | 100 | 88 | 64 | 97 |
| Livermore-7 | 0 | 4 | 0 | 0 | 0 |
| Livermore-8 | 0 | 0 | 0 | 0 | 0 |
| Livermore-9 | 4 | 88 | 24 | 12 | 0 |
| Livermore-10 | 100 | 8 | 24 | 0 | 0 |
| Livermore-11 | 100 | 100 | 100 | 92 | 17 |
| Livermore-12 | 100 | 100 | 100 | 100 | 61 |
| Livermore-13 | 100 | 100 | 100 | 84 | 55 |
| Livermore-14 | 100 | 100 | 100 | 100 | 0 |
| Livermore-15 | 100 | 100 | 100 | 96 | 0 |
| Livermore-16 | 100 | 100 | 92 | 12 | 4 |
| Livermore-17 | 100 | 36 | 68 | 4 | 0 |
| Livermore-18 | 100 | 48 | 56 | 0 | 0 |
| Livermore-19 | 100 | 100 | 100 | 100 | 100 |
| Livermore-20 | 100 | 100 | 100 | 100 | 98 |
| Livermore-21 | 76 | 88 | 24 | 64 | 2 |
| Livermore-22 | 100 | 92 | 84 | 68 | 3 |
| **Livermore Average** | **85.45** | 77.45 | 71.09 | 56.36 | 30.41 |
| **All Average** | **90.59** | 82.59 | 78.59 | 66.82 | 49.18 |

Table 11: Runtimes of GSR vs. NGGPPS on the Nguyen benchmarks. The ground truth expressions for these benchmarks are shown in Table 19.

|  | Runtime (sec) | |
| --- | --- | --- |
| **Benchmark** | **GSR** | **NGGPPS** |
| Nguyen-1 | 18.66 | 27.05 |
| Nguyen-2 | 25.96 | 59.79 |
| Nguyen-3 | 38.77 | 151.06 |
| Nguyen-4 | 63.82 | 268.88 |
| Nguyen-5 | 447.01 | 501.65 |
| Nguyen-6 | 465.79 | 43.96 |
| Nguyen-7 | 33.35 | 752.32 |
| Nguyen-8 | 93.89 | 123.21 |
| Nguyen-9 | 391.79 | 31.17 |
| Nguyen-10 | 68.05 | 103.72 |
| Nguyen-11 | 38.86 | 66.50 |
| **Average** | **153.27** | 193.57 |

Table 12: Average performance in mean RMSE and runtime, along with their standard errors, for GSR against s-GSR and several strong SR methods on the SymSet benchmark problem sets over 25 independent runs. The ground truth expressions for these benchmarks are shown in Table 20.

| Benchmark | Mean RMSE | | Runtime (sec) | |
|---|---|---|---|---|
| | **GSR** | **s-GSR** | **GSR** | **s-GSR** |
| SymSet-1 | $2.52 \times 10^{-5} \pm 9.29 \times 10^{-6}$ | $3.69 \times 10^{-5} \pm 1.62 \times 10^{-5}$ | $49.94 \pm 2.61$ | $43.81 \pm 1.95$ |
| SymSet-2 | $4.28 \times 10^{-7} \pm 3.35 \times 10^{-8}$ | $2.34 \times 10^{-3} \pm 1.21 \times 10^{-3}$ | $75.24 \pm 2.83$ | $91.95 \pm 4.26$ |
| SymSet-3 | $6.58 \times 10^{-6} \pm 1.79 \times 10^{-6}$ | $1.22 \times 10^{-3} \pm 8.17 \times 10^{-4}$ | $66.34 \pm 2.51$ | $88.14 \pm 3.49$ |
| SymSet-4 | $7.94 \times 10^{-4} \pm 5.19 \times 10^{-4}$ | $1.37 \times 10^{-4} \pm 1.46 \times 10^{-3}$ | $428.23 \pm 8.46$ | $405.54 \pm 6.04$ |
| SymSet-5 | $3.63 \times 10^{-5} \pm 9.92 \times 10^{-5}$ | $4.98 \times 10^{-5} \pm 7.37 \times 10^{-5}$ | $71.86 \pm 4.12$ | $64.19 \pm 3.82$ |
| SymSet-6 | $4.38 \times 10^{-5} \pm 8.09 \times 10^{-6}$ | $8.12 \times 10^{-2} \pm 1.93 \times 10^{-2}$ | $76.59 \pm 3.72$ | $94.18 \pm 4.16$ |
| SymSet-7 | $2.39 \times 10^{-4} \pm 1.71 \times 10^{-5}$ | $6.83 \times 10^{-2} \pm 6.56 \times 10^{-3}$ | $93.61 \pm 2.54$ | $109.94 \pm 4.98$ |
| SymSet-8 | $6.83 \times 10^{-4} \pm 1.95 \times 10^{-5}$ | $7.88 \times 10^{-3} \pm 3.88 \times 10^{-3}$ | $68.07 \pm 2.47$ | $90.61 \pm 3.06$ |
| SymSet-9 | $3.57 \times 10^{-6} \pm 3.41 \times 10^{-5}$ | $6.32 \times 10^{-3} \pm 1.39 \times 10^{-3}$ | $57.36 \pm 2.68$ | $83.18 \pm 2.09$ |
| SymSet-10 | $2.12 \times 10^{-4} \pm 4.43 \times 10^{-4}$ | $3.24 \times 10^{-4} \pm 7.58 \times 10^{-5}$ | $393.62 \pm 5.47$ | $379.56 \pm 6.86$ |
| SymSet-11 | $1.43 \times 10^{-3} \pm 9.97 \times 10^{-4}$ | $9.39 \times 10^{-2} \pm 6.78 \times 10^{-3}$ | $124.38 \pm 9.65$ | $187.49 \pm 8.35$ |
| SymSet-12 | $1.22 \times 10^{-5} \pm 5.37 \times 10^{-5}$ | $7.09 \times 10^{-5} \pm 2.88 \times 10^{-4}$ | $78.53 \pm 4.02$ | $69.72 \pm 2.97$ |
| SymSet-13 | $2.31 \times 10^{-5} \pm 1.83 \times 10^{-5}$ | $9.13 \times 10^{-2} \pm 3.28 \times 10^{-2}$ | $96.14 \pm 5.48$ | $114.85 \pm 4.61$ |
| SymSet-14 | $2.18 \times 10^{-5} \pm 9.47 \times 10^{-6}$ | $6.14 \times 10^{-2} \pm 7.78 \times 10^{-3}$ | $112.32 \pm 4.57$ | $137.61 \pm 4.53$ |
| SymSet-15 | $1.81 \times 10^{-6} \pm 4.75 \times 10^{-5}$ | $6.71 \times 10^{-3} \pm 3.57 \times 10^{-4}$ | $46.97 \pm 2.33$ | $85.07 \pm 4.14$ |
| SymSet-16 | $7.24 \times 10^{-4} \pm 3.58 \times 10^{-4}$ | $9.86 \times 10^{-3} \pm 2.19 \times 10^{-3}$ | $98.37 \pm 3.71$ | $126.16 \pm 5.94$ |
| SymSet-17 | $2.57 \times 10^{-4} \pm 7.03 \times 10^{-5}$ | $3.41 \times 10^{-3} \pm 4.54 \times 10^{-3}$ | $116.78 \pm 4.49$ | $143.31 \pm 6.28$ |
| **Average** | $\mathbf{2.66 \times 10^{-4} \pm 1.59 \times 10^{-4}}$ | $2.56 \times 10^{-2} \pm 5.27 \times 10^{-3}$ | $120.84 \pm 4.22$ | $136.19 \pm 4.56$ |
| Benchmark | **NGGPPS** | **PSTree** | **NGGPPS** | **PSTree** |
| SymSet-1 | $3.66 \times 10^{-1} \pm 7.81 \times 10^{-3}$ | $7.92 \times 10^{-3} \pm 2.23 \times 10^{-3}$ | $175.32 \pm 1.54$ | $42.98 \pm 3.23$ |
| SymSet-2 | $4.75 \times 10^{-1} \pm 5.92 \times 10^{-2}$ | $1.96 \times 10^{-1} \pm 2.72 \times 10^{-2}$ | $171.46 \pm 2.03$ | $24.68 \pm 1.05$ |
| SymSet-3 | $1.34 \times 10^{-2} \pm 1.91 \times 10^{-3}$ | $3.73 \times 10^{-3} \pm 4.01 \times 10^{-4}$ | $167.11 \pm 1.69$ | $21.68 \pm 1.89$ |
| SymSet-4 | $1.76 \times 10^{0} \pm 9.89 \times 10^{-1}$ | $1.32 \times 10^{0} \pm 1.61 \times 10^{-1}$ | $171.85 \pm 2.01$ | $24.14 \pm 1.09$ |
| SymSet-5 | $4.21 \times 10^{-1} \pm 3.58 \times 10^{-2}$ | $3.19 \times 10^{-1} \pm 9.94 \times 10^{-2}$ | $177.98 \pm 1.57$ | $13.09 \pm 0.37$ |
| SymSet-6 | $2.08 \times 10^{0} \pm 4.80 \times 10^{-1}$ | $1.42 \times 10^{0} \pm 3.72 \times 10^{-1}$ | $179.34 \pm 1.75$ | $13.28 \pm 0.36$ |
| SymSet-7 | $2.43 \times 10^{-2} \pm 9.62 \times 10^{-3}$ | $5.25 \times 10^{-1} \pm 7.62 \times 10^{-2}$ | $133.89 \pm 6.95$ | $11.78 \pm 0.24$ |
| SymSet-8 | $3.93 \times 10^{-1} \pm 5.08 \times 10^{-2}$ | $4.02 \times 10^{-1} \pm 6.42 \times 10^{-2}$ | $169.46 \pm 1.81$ | $11.54 \pm 0.41$ |
| SymSet-9 | $1.34 \times 10^{-1} \pm 3.55 \times 10^{-2}$ | $1.23 \times 10^{-1} \pm 1.90 \times 10^{-2}$ | $175.04 \pm 1.04$ | $12.23 \pm 0.29$ |
| SymSet-10 | $3.32 \times 10^{-1} \pm 3.09 \times 10^{-2}$ | $9.91 \times 10^{-1} \pm 9.59 \times 10^{-2}$ | $178.27 \pm 1.75$ | $11.06 \pm 0.17$ |
| SymSet-11 | $2.49 \times 10^{-1} \pm 2.71 \times 10^{-2}$ | $1.11 \times 10^{-1} \pm 2.24 \times 10^{-2}$ | $166.91 \pm 1.66$ | $21.26 \pm 0.53$ |
| SymSet-12 | $8.76 \times 10^{-1} \pm 6.89 \times 10^{-2}$ | $8.32 \times 10^{-1} \pm 1.39 \times 10^{-1}$ | $178.56 \pm 1.53$ | $11.65 \pm 0.32$ |
| SymSet-13 | $2.19 \times 10^{-1} \pm 1.77 \times 10^{-1}$ | $8.69 \times 10^{-1} \pm 1.71 \times 10^{-1}$ | $126.39 \pm 7.88$ | $9.92 \pm 0.23$ |
| SymSet-14 | $4.93 \times 10^{-17} \pm 2.85 \times 10^{-18}$ | $8.24 \times 10^{-2} \pm 8.50 \times 10^{-3}$ | $99.85 \pm 3.81$ | $9.95 \pm 0.21$ |
| SymSet-15 | $2.86 \times 10^{-17} \pm 4.49 \times 10^{-18}$ | $6.22 \times 10^{-2} \pm 1.28 \times 10^{-2}$ | $94.42 \pm 2.84$ | $9.93 \pm 0.18$ |
| SymSet-17 | $9.16 \times 10^{-2} \pm 6.52 \times 10^{-3}$ | $5.44 \times 10^{-2} \pm 5.51 \times 10^{-3}$ | $171.25 \pm 1.55$ | $10.46 \pm 0.19$ |
| **Average** | $4.65 \times 10^{-1} \pm 1.24 \times 10^{-1}$ | $4.57 \times 10^{-1} \pm 7.98 \times 10^{-2}$ | $158.57 \pm 2.59$ | $\mathbf{16.23 \pm 0.67}$ |
| Benchmark | **PySR** | **gplearn** | **PySR** | **gplearn** |
| SymSet-1 | $1.10 \times 10^{-3} \pm 3.01 \times 10^{-4}$ | $4.45 \times 10^{-2} \pm 2.91 \times 10^{-3}$ | $61.21 \pm 19.72$ | $140.44 \pm 7.27$ |
| SymSet-2 | $5.75 \times 10^{-2} \pm 2.28 \times 10^{-2}$ | $3.42 \times 10^{-1} \pm 5.12 \times 10^{-2}$ | $119.55 \pm 39.39$ | $145.84 \pm 1.67$ |
| SymSet-3 | $1.71 \times 10^{-3} \pm 2.02 \times 10^{-4}$ | $1.72 \times 10^{-2} \pm 9.41 \times 10^{-3}$ | $12.62 \pm 0.69$ | $169.38 \pm 1.19$ |
| SymSet-4 | $6.14 \times 10^{-1} \pm 6.12 \times 10^{-1}$ | $2.89 \times 10^{0} \pm 5.77 \times 10^{-1}$ | $79.39 \pm 1.98$ | $248.12 \pm 9.71$ |
| SymSet-5 | $5.91 \times 10^{-2} \pm 1.28 \times 10^{-2}$ | $2.64 \times 10^{-1} \pm 1.76 \times 10^{-2}$ | $69.47 \pm 0.49$ | $190.83 \pm 1.41$ |
| SymSet-6 | $6.57 \times 10^{0} \pm 1.96 \times 10^{0}$ | $3.94 \times 10^{0} \pm 1.31 \times 10^{0}$ | $67.45 \pm 0.67$ | $215.11 \pm 6.69$ |
| SymSet-7 | $5.22 \times 10^{-2} \pm 4.64 \times 10^{-2}$ | $8.93 \times 10^{-1} \pm 1.50 \times 10^{-1}$ | $117.47 \pm 28.85$ | $169.94 \pm 1.91$ |
| SymSet-8 | $3.57 \times 10^{-1} \pm 3.85 \times 10^{-2}$ | $4.35 \times 10^{-1} \pm 3.16 \times 10^{-2}$ | $135.13 \pm 30.82$ | $167.64 \pm 0.98$ |
| SymSet-9 | $2.58 \times 10^{-2} \pm 8.51 \times 10^{-3}$ | $2.19 \times 10^{-1} \pm 4.17 \times 10^{-2}$ | $162.11 \pm 45.01$ | $159.22 \pm 1.86$ |
| SymSet-10 | $4.14 \times 10^{-2} \pm 1.72 \times 10^{-2}$ | $1.63 \times 10^{0} \pm 3.38 \times 10^{-1}$ | $48.17 \pm 0.43$ | $192.92 \pm 1.72$ |
| SymSet-11 | $2.81 \times 10^{-2} \pm 3.40 \times 10^{-3}$ | $1.88 \times 10^{-1} \pm 2.91 \times 10^{-2}$ | $49.88 \pm 0.63$ | $184.38 \pm 1.34$ |
| SymSet-12 | $2.53 \times 10^{-2} \pm 1.11 \times 10^{-2}$ | $2.62 \times 10^{-1} \pm 3.11 \times 10^{-2}$ | $136.14 \pm 41.77$ | $187.32 \pm 2.90$ |
| SymSet-13 | $8.44 \times 10^{-2} \pm 6.39 \times 10^{-2}$ | $1.07 \times 10^{-16} \pm 1.29 \times 10^{-17}$ | $16.01 \pm 2.98$ | $12.52 \pm 0.77$ |
| SymSet-14 | $1.59 \times 10^{-2} \pm 4.51 \times 10^{-3}$ | $9.82 \times 10^{-2} \pm 1.59 \times 10^{-2}$ | $57.54 \pm 9.81$ | $142.12 \pm 5.79$ |
| SymSet-15 | $6.90 \times 10^{-3} \pm 4.02 \times 10^{-3}$ | $2.05 \times 10^{-1} \pm 1.36 \times 10^{-2}$ | $93.54 \pm 56.54$ | $147.58 \pm 1.12$ |
| SymSet-17 | $3.94 \times 10^{-2} \pm 5.21 \times 10^{-3}$ | $1.18 \times 10^{-1} \pm 8.20 \times 10^{-3}$ | $167.45 \pm 65.81$ | $148.42 \pm 0.67$ |
| **Average** | $4.99 \times 10^{-1} \pm 1.76 \times 10^{-1}$ | $7.22 \times 10^{-1} \pm 1.64 \times 10^{-1}$ | $87.07 \pm 21.60$ | $163.86 \pm 2.94$ |

# B  More Examples on Our Matrix-Based Encoding Scheme

The encoding process happens according to a table of mapping rules that is very straightforward to understand and use. For example, consider the mapping rules shown in Table 13 below, where $d$ is the dimension of the input feature vector. Note that Table 13 involves more transformations than Table 2.

Table 13: An example table of mapping rules for a basis function. Placeholder operands are denoted by $\bullet$, e.g. $\bullet^2$ corresponds to the square operator. The identity operator is denoted by $\bullet^1$.

| $b_{\bullet,1}$ Transformation $(T)$ | 0 | 1 | 2 | 3 | 4 | 5 | 6 | 7 | 8 | 9 |
|---|---|---|---|---|---|---|---|---|---|---|
| | 1 | $\bullet^1$ | $\bullet^{-1}$ | $\bullet^2$ | $\bullet^3$ | cos | sin | exp | ln | $\sqrt{\bullet}$ |

| $b_{\bullet,2}$ Argument Type $(arg)$ | 0 | 1 | 2 |
|---|---|---|---|
| | $x$ | $\sum$ | $\prod$ |

| $b_{\bullet,3}, \cdots, b_{\bullet,m_{\boldsymbol{B}}}$ Variable $(v)$ | 0 | 1 | 2 | 3 | $\cdots$ | $d$ |
|---|---|---|---|---|---|---|
| | skip | $x_1$ | $x_2$ | $x_3$ | $\cdots$ | $x_d$ |

**Example 1.** For $d = 2$, $n_{\boldsymbol{B}^\phi} = 2$ and $m_{\boldsymbol{B}^\phi} = 4$ (i.e. $n_v = 2$), the basis function $\phi(\mathbf{x}) = x_1^2 e^{x_1 x_2}$ can be generated according to the encoding steps shown in Table 14.

Table 14: Encoding steps corresponding to the basis function $\phi(\mathbf{x}) = x_1^2 e^{x_1 x_2}$.

| Step | $T$ | $arg$ | $v_1$ | $v_2$ | Update |
|---|---|---|---|---|---|
| 1 | $\bullet^2$ | $x$ | $x_1$ | — | $T_1(\mathbf{x}) = x_1^2$ |
| 2 | exp | $\prod$ | $x_1$ | $x_2$ | $T_2(\mathbf{x}) = e^{x_1 x_2}$ |
| Final Update: $\phi(\mathbf{x}) = T_1(\mathbf{x}) \cdot T_2(\mathbf{x})$ | | | | | |

Based on the mapping rules in Table 13 and the encoding steps in Table 14, the basis function $\phi(\mathbf{x}) = x_1^2 e^{x_1 x_2}$ can be encoded by a $2 \times 4$ matrix as follows:

$$\boldsymbol{B}^\phi = \begin{bmatrix} 3 & 0 & 1 & \bullet \\ 7 & 2 & 1 & 2 \end{bmatrix} \tag{21}$$

**Example 2.** For $d = 3$, $n_{\boldsymbol{B}^\phi} = 5$ and $m_{\boldsymbol{B}^\phi} = 5$ (i.e. $n_v = 3$), the basis function $\phi(\mathbf{x}) = \frac{x_2^3 \sin(x_2 x_3) \sqrt{x_2 + 2x_3}}{2x_1 + x_2}$ can be generated according to the encoding steps shown in Table 15.

Table 15: Encoding steps corresponding to the basis function $\phi(\mathbf{x}) = \frac{x_2^3 \sin(x_2 x_3) \sqrt{x_2 + 2x_3}}{2x_1 + x_2}$.

| Step | $T$ | $arg$ | $v_1$ | $v_2$ | $v_3$ | Update |
|---|---|---|---|---|---|---|
| 1 | $\bullet^{-1}$ | $\sum$ | $x_1$ | $x_1$ | $x_2$ | $T_1(\mathbf{x}) = (2x_1 + x_2)^{-1}$ |
| 2 | $\bullet^3$ | $x$ | $x_2$ | — | — | $T_2(\mathbf{x}) = x_2^3$ |
| 3 | sin | $\prod$ | $x_2$ | $x_3$ | — | $T_3(\mathbf{x}) = \sin(x_2 x_3)$ |
| 4 | $\sqrt{\bullet}$ | $\sum$ | $x_2$ | $x_3$ | $x_3$ | $T_4(\mathbf{x}) = \sqrt{x_2 + 2x_3}$ |
| 5 | 1 | — | — | — | — | $T_5(\mathbf{x}) = 1$ |
| Final Update: $\phi(\mathbf{x}) = T_1(\mathbf{x}) \cdot T_2(\mathbf{x}) \cdot T_3(\mathbf{x}) \cdot T_4(\mathbf{x}) \cdot T_5(\mathbf{x})$ | | | | | | |

Based on the mapping rules in Table 13 and the encoding steps in Table 15, the basis function $\phi(\mathbf{x}) = \frac{x_2^3 \sin(x_2 x_3) \sqrt{x_2 + 2x_3}}{2x_1 + x_2}$ can be encoded by a $5 \times 5$ matrix as follows:

$$\boldsymbol{B}^\phi = \begin{bmatrix} 2 & 1 & 1 & 1 & 2 \\ 4 & 0 & 2 & \bullet & \bullet \\ 6 & 2 & 2 & 3 & \bullet \\ 9 & 1 & 2 & 3 & 3 \\ 0 & \bullet & \bullet & \bullet & \bullet \end{bmatrix} \tag{22}$$

**Example 3.** For $n_{\boldsymbol{B}^\psi} = 2$, the basis function $\psi(y) = y^3\sqrt{y}$ can be generated according to the encoding steps shown in Table 16.

Table 16: Encoding steps corresponding to the basis function $\psi(y) = y^3\sqrt{y}$.

| Step | $T$ | Update |
|------|-----|--------|
| 1 | $\bullet^3$ | $T_1(y) = y^3$ |
| 2 | $\sqrt{\bullet}$ | $T_2(y) = \sqrt{y}$ |
| Final Update: $\psi(y) = T_1(y) \cdot T_2(y)$ | | |

Based on the mapping rules in Table 13 and the encoding steps in Table 17, the basis function $\psi(y) = y^3\sqrt{y}$ can be encoded by a $2 \times 1$ matrix as follows:

$$\boldsymbol{B}^\psi = \begin{bmatrix} 4 \\ 9 \end{bmatrix} \tag{23}$$

**Example 4.** For $n_{\boldsymbol{B}^\psi} = 3$, the basis function $\psi(y) = \ln(y)$ can be generated according to the encoding steps shown in Table 17.

Table 17: Encoding steps corresponding to the basis function $\psi(y) = \ln(y)$.

| Step | $T$ | Update |
|------|-----|--------|
| 1 | 1 | $T_1(y) = 1$ |
| 2 | ln | $T_2(y) = \ln(y)$ |
| 3 | 1 | $T_3(y) = 1$ |
| Final Update: $\psi(y) = T_1(y) \cdot T_2(y) \cdot T_3(y)$ | | |

Based on the mapping rules in Table 13 and the encoding steps in Table 17, the basis function $\psi(y) = \ln(y)$ can be encoded by a $3 \times 1$ matrix as follows:

$$\boldsymbol{B}^\psi = \begin{bmatrix} 0 \\ 8 \\ 0 \end{bmatrix} \tag{24}$$

**Example 5.** For $n_{\boldsymbol{B}^\psi} = 1$, the basis function $\psi(y) = e^y$ can be generated according to the encoding steps shown in Table 18.

Table 18: Encoding steps corresponding to the basis function $\psi(y) = e^y$.

| Step | $T$ | Update |
|------|-----|--------|
| 1 | exp | $T_1(y) = e^y$ |
| Final Update: $\psi(y) = T_1(y)$ | | |

Based on the mapping rules in Table 13 and the encoding steps in Table 18, the basis function $\psi(y) = e^y$ can be encoded by a $1 \times 1$ matrix as follows:

$$\boldsymbol{B}^\psi = \begin{bmatrix} 7 \end{bmatrix} \tag{25}$$

## C   Symbolic Regression Benchmark Problem Sets

Table 19: Specifications of the Symbolic Regression (SR) benchmark problems. Input variables are denoted by $x$ for 1-dimensional problems, and by $(x_1, x_2)$ for 2-dimensional problems. $U(a, b, c)$ indicates $c$ random points uniformly sampled between $a$ and $b$ for every input variable; different random seeds are used for the training and test sets. $E(a, b, c)$ indicates $c$ evenly spaced points between $a$ and $b$ for every input variable; the same points are used for the training and test sets (except Neat-6, which uses $E(1, 120, 120)$ as test set, and the Jin tests, which use $U(-3, 3, 30)$ as test set). To simplify the notation, libraries (of allowable arithmetic operators and mathematical functions) are defined relative to a 'base' library $\mathcal{L}_0 = \{+, -, \times, \div, \cos, \sin, \exp, \ln\}$. Placeholder operands are denoted by $\bullet$, e.g. $\bullet^2$ corresponds to the square operator.

| Benchmark | Expression | Dataset | Library |
|---|---|---|---|
| Nguyen-1 | $y = x^3 + x^2 + x$ | $U(-1, 1, 20)$ | $\mathcal{L}_0$ |
| Nguyen-2 | $y = x^4 + x^3 + x^2 + x$ | $U(-1, 1, 20)$ | $\mathcal{L}_0$ |
| Nguyen-3 | $y = x^5 + x^4 + x^3 + x^2 + x$ | $U(-1, 1, 20)$ | $\mathcal{L}_0$ |
| Nguyen-4 | $y = x^6 + x^5 + x^4 + x^3 + x^2 + x$ | $U(-1, 1, 20)$ | $\mathcal{L}_0$ |
| Nguyen-5 | $y = \sin(x^2)\cos(x) - 1$ | $U(-1, 1, 20)$ | $\mathcal{L}_0$ |
| Nguyen-6 | $y = \sin(x) + \sin(x + x^2)$ | $U(-1, 1, 20)$ | $\mathcal{L}_0$ |
| Nguyen-7 | $y = \ln(x + 1) + \ln(x^2 + 1)$ | $U(0, 2, 20)$ | $\mathcal{L}_0$ |
| Nguyen-8 | $y = \sqrt{x}$ | $U(0, 4, 20)$ | $\mathcal{L}_0$ |
| Nguyen-9 | $y = \sin(x_1) + \sin(x_2^2)$ | $U(0, 1, 20)$ | $\mathcal{L}_0$ |
| Nguyen-10 | $y = 2\sin(x_1)\cos(x_2)$ | $U(0, 1, 20)$ | $\mathcal{L}_0$ |
| Nguyen-11 | $y = x_1^{x_2}$ | $U(0, 1, 20)$ | $\mathcal{L}_0$ |
| Nguyen-12 | $y = x_1^4 - x_1^3 + \frac{1}{2}x_2^2 - x_2$ | $U(0, 1, 20)$ | $\mathcal{L}_0$ |
| Nguyen-12$^\star$ | $y = x_1^4 - x_1^3 + \frac{1}{2}x_2^2 - x_2$ | $U(0, 10, 20)$ | $\mathcal{L}_0$ |
| Jin-1 | $y = 2.5x_1^4 - 1.3x_1^3 + 0.5x_2^2 - 1.7x_2$ | $U(-3, 3, 100)$ | $\mathcal{L}_0 - \{\ln\} \cup \{\bullet^2, \bullet^3, \text{const}\}$ |
| Jin-2 | $y = 8x_1^2 + 8x_2^3 - 15$ | $U(-3, 3, 100)$ | $\mathcal{L}_0 - \{\ln\} \cup \{\bullet^2, \bullet^3, \text{const}\}$ |
| Jin-3 | $y = 0.2x_1^3 + 0.5x_2^3 - 1.2x_2 - 0.5x_1$ | $U(-3, 3, 100)$ | $\mathcal{L}_0 - \{\ln\} \cup \{\bullet^2, \bullet^3, \text{const}\}$ |
| Jin-4 | $y = 1.5e^{x_1} + 5\cos(x_2)$ | $U(-3, 3, 100)$ | $\mathcal{L}_0 - \{\ln\} \cup \{\bullet^2, \bullet^3, \text{const}\}$ |
| Jin-5 | $y = 6\sin(x_1)\cos(x_2)$ | $U(-3, 3, 100)$ | $\mathcal{L}_0 - \{\ln\} \cup \{\bullet^2, \bullet^3, \text{const}\}$ |
| Jin-6 | $y = 1.35x_1x_2 + 5.5\sin\left((x_1 - 1)(x_2 - 1)\right)$ | $U(-3, 3, 100)$ | $\mathcal{L}_0 - \{\ln\} \cup \{\bullet^2, \bullet^3, \text{const}\}$ |
| Neat-1 | $y = x^4 + x^3 + x^2 + x$ | $U(-1, 1, 20)$ | $\mathcal{L}_0 \cup \{1\}$ |
| Neat-2 | $y = x^5 + x^4 + x^3 + x^2 + x$ | $U(-1, 1, 20)$ | $\mathcal{L}_0 \cup \{1\}$ |
| Neat-3 | $y = \sin(x^2)\cos(x) - 1$ | $U(-1, 1, 20)$ | $\mathcal{L}_0 \cup \{1\}$ |
| Neat-4 | $y = \ln(x + 1) + \ln(x^2 + 1)$ | $U(0, 2, 20)$ | $\mathcal{L}_0 \cup \{1\}$ |
| Neat-5 | $y = 2\sin(x_1)\cos(x_2)$ | $U(-1, 1, 100)$ | $\mathcal{L}_0$ |
| Neat-6 | $y = \sum_{k=1}^{x} \frac{1}{k}$ | $E(1, 50, 50)$ | $\{+, \times, \div, \bullet^{-1}, -\bullet, \sqrt{\bullet}\}$ |
| Neat-7 | $y = 2 - 2.1\cos(9.8x_1)\sin(1.3x_2)$ | $E(-50, 50, 10^5)$ | $\mathcal{L}_0 \cup \{\tan, \tanh, \bullet^2, \bullet^3, \sqrt{\bullet}\}$ |
| Neat-8 | $y = \frac{e^{-(x_1-1)^2}}{1.2 + (x_2 - 2.5)^2}$ | $U(0.3, 4, 100)$ | $\{+, -, \times, \div, \exp, e^{-\bullet}, \bullet^2\}$ |
| Neat-9 | $y = \frac{1}{1 + x_1^{-4}} + \frac{1}{1 + x_2^{-4}}$ | $E(-5, 5, 21)$ | $\mathcal{L}_0$ |

Table 20: Specifications of the Symbolic Regression (SR) benchmark problems. Input variables are denoted by $x$ for 1-dimensional problems, by $(x_1, x_2)$ for 2-dimensional problems, and by $(x_1, x_2, x_3)$ for 3-dimensional problems. $U(a, b, c)$ indicates $c$ random points uniformly sampled between $a$ and $b$ for every input variable; different random seeds are used for the training and test sets. To simplify the notation, libraries (of allowable arithmetic operators and mathematical functions) are defined relative to 'base' libraries $\mathcal{L}_0 = \{+, -, \times, \div, \cos, \sin, \exp, \ln\}$ or $\mathcal{L}_0^c = \mathcal{L}_0 \cup \{\text{const}\}$. Placeholder operands are denoted by $\bullet$, e.g. $\bullet^2$ corresponds to the square operator.

| Benchmark | Expression | Dataset | Library |
|---|---|---|---|
| Livermore-1 | $y = \frac{1}{3} + x + \sin(x^2)$ | $U(-10, 10, 1000)$ | $\mathcal{L}_0$ |
| Livermore-2 | $y = \sin(x^2)\cos(x) - 2$ | $U(-1, 1, 20)$ | $\mathcal{L}_0$ |
| Livermore-3 | $y = \sin(x^3)\cos(x^2) - 1$ | $U(-1, 1, 20)$ | $\mathcal{L}_0$ |
| Livermore-4 | $y = \ln(x+1) + \ln(x^2+1) + \ln(x)$ | $U(0, 2, 20)$ | $\mathcal{L}_0$ |
| Livermore-5 | $y = x_1^4 - x_1^3 + x_1^2 - x_2$ | $U(0, 1, 20)$ | $\mathcal{L}_0$ |
| Livermore-6 | $y = 4x^4 + 3x^3 + 2x^2 + x$ | $U(-1, 1, 20)$ | $\mathcal{L}_0$ |
| Livermore-7 | $y = \sinh(x)$ | $U(-1, 1, 20)$ | $\mathcal{L}_0$ |
| Livermore-8 | $y = \cosh(x)$ | $U(-1, 1, 20)$ | $\mathcal{L}_0$ |
| Livermore-9 | $y = x^9 + x^8 + x^7 + x^6 + x^5 + x^4 + x^3 + x^2 + x$ | $U(-1, 1, 20)$ | $\mathcal{L}_0$ |
| Livermore-10 | $y = 6\sin(x_1)\cos(x_2)$ | $U(0, 1, 20)$ | $\mathcal{L}_0$ |
| Livermore-11 | $y = \frac{x_1^2 x_1^2}{x_1 + x_2}$ | $U(-1, 1, 50)$ | $\mathcal{L}_0$ |
| Livermore-12 | $y = \frac{x_1^5}{x_2^3}$ | $U(-1, 1, 50)$ | $\mathcal{L}_0$ |
| Livermore-13 | $y = x^{\frac{1}{3}}$ | $U(0, 4, 20)$ | $\mathcal{L}_0$ |
| Livermore-14 | $y = x^3 + x^2 + x + \sin(x) + \sin(x^2)$ | $U(-1, 1, 20)$ | $\mathcal{L}_0$ |
| Livermore-15 | $y = x^{\frac{1}{5}}$ | $U(0, 4, 20)$ | $\mathcal{L}_0$ |
| Livermore-16 | $y = x^{\frac{2}{5}}$ | $U(0, 4, 20)$ | $\mathcal{L}_0$ |
| Livermore-17 | $y = 4\sin(x_1)\cos(x_2)$ | $U(0, 1, 20)$ | $\mathcal{L}_0$ |
| Livermore-18 | $y = \sin(x^2)\cos(x) - 5$ | $U(-1, 1, 20)$ | $\mathcal{L}_0$ |
| Livermore-19 | $y = x^5 + x^4 + x^2 + x$ | $U(-1, 1, 20)$ | $\mathcal{L}_0$ |
| Livermore-20 | $y = e^{-x^2}$ | $U(-1, 1, 20)$ | $\mathcal{L}_0$ |
| Livermore-21 | $y = x^8 + x^7 + x^6 + x^5 + x^4 + x^3 + x^2 + x$ | $U(-1, 1, 20)$ | $\mathcal{L}_0$ |
| Livermore-22 | $y = e^{-0.5x^2}$ | $U(-1, 1, 20)$ | $\mathcal{L}_0$ |
| SymSet-1 | $y = x\sinh(x) - \frac{4}{5}$ | $U(-1, 1, 20)$ | $\mathcal{L}_0^c - \{\ln\} \cup \{e^{-\bullet}\}$ |
| SymSet-2 | $y = (x^5 - 3x^4 - 2.8x + 5)^{-1}$ | $U(-1, 1, 20)$ | $\mathcal{L}_0^c \cup \{\bullet^{-1}\}$ |
| SymSet-3 | $y = (x^4 - 1.2x^2 + 11.5)^{\frac{1}{3}}$ | $U(-1, 1, 20)$ | $\mathcal{L}_0^c \cup \{\bullet^2, \bullet^3\}$ |
| SymSet-4 | $y = 0.8 - \cos(x) + 4.2e^x \sin(x^2)$ | $U(-3, 3, 20)$ | $\mathcal{L}_0^c$ |
| SymSet-5 | $y = 4.5x_1^2 + x_1 x_2^3 - 1.7x_2 - 3.1$ | $U(-1, 1, 20)$ | $\mathcal{L}_0^c$ |
| SymSet-6 | $y = \frac{5}{3x_1 - x_2^3}$ | $U(-1, 1, 20)$ | $\mathcal{L}_0^c \cup \{\bullet^{-1}\}$ |
| SymSet-7 | $y = \ln(x_1^3 + 4x_1 x_2)$ | $U(0, 2, 20)$ | $\mathcal{L}_0^c$ |
| SymSet-8 | $y = \sqrt{5x_1^5 + 14x_1^3 x_2^4 - 2x_2 + 7}$ | $U(-1, 1, 20)$ | $\mathcal{L}_0^c \cup \{\bullet^2, \bullet^3\}$ |
| SymSet-9 | $y = (2x_1 + x_2)^{-\frac{2}{3}}$ | $U(0, 2, 20)$ | $\mathcal{L}_0^c$ |
| SymSet-10 | $y = 1.5\cos(x_1)\ln(x_1 x_2) - 2.5$ | $U(0, 1, 20)$ | $\mathcal{L}_0^c$ |
| SymSet-11 | $y = \sqrt{2\cos(x_1) + 30e^{x_2}} + 4$ | $U(-1, 1, 20)$ | $\mathcal{L}_0^c \cup \{\bullet^2\}$ |
| SymSet-12 | $y = 0.4x_1^4 + 6.2x_2 - 3.5x_1 x_3 - 4.5$ | $U(-1, 1, 20)$ | $\mathcal{L}_0^c$ |
| SymSet-13 | $y = \frac{2x_2}{x_1 + x_3}$ | $U(0, 1, 20)$ | $\mathcal{L}_0^c$ |
| SymSet-14 | $y = \frac{x_1 x_2 x_3}{x_1 + x_2 + x_3}$ | $U(0, 2, 20)$ | $\mathcal{L}_0^c$ |
| SymSet-15 | $y = (x_1 + x_2)^{x_3}$ | $U(0, 1, 20)$ | $\mathcal{L}_0^c$ |
| SymSet-16 | $y = e^{2.6x_1 - \ln(x_2) + 9.8\cos(x_3)}$ | $U(0, 1, 20)$ | $\mathcal{L}_0^c$ |
| SymSet-17 | $y = \ln\left(0.2e^{x_1 + x_2} + 0.5\cos(x_3^2)\right)$ | $U(0, 1, 20)$ | $\mathcal{L}_0^c$ |

## D   Typical Recovered Expressions

Table 21: Typical expressions (with exact symbolic equivalence) recovered by GSR for the Nguyen benchmark set. Note that the coefficients in the GSR expressions form a unit vector due to the normalization constraint imposed by the Lasso problem in Equation 6. It is easy to verify (by simplification) that the GSR expressions are symbolically equivalent to the ground truth expressions.

| Benchmark | | Expression |
|---|---|---|
| Nguyen-1 | Truth | $y = x^3 + x^2 + x$ |
| | GSR | $-0.558\,y = -0.1395(x \times x \times x) - 0.1395(x \times x \times x) - 0.2697x - 0.1395(x \times x \times x)$ $-0.2697x - 0.2697x + 0.0651(x + x + x + x) + 0.0651(x + x + x + x)$ $-0.1395(x \times x \times x) - 0.2697x - 0.558(x \times x)$ |
| Nguyen-2 | Truth | $y = x^4 + x^3 + x^2 + x$ |
| | GSR | $-0.58554\,y = -0.09759x(x + x) - 0.58554x - 0.19518(x \times x)x - 0.09759(x + x)x$ $-0.19518(x \times x \times x) - 0.29277x(x \times x \times x) - 0.29277x(x \times x \times x)$ $-0.09759(x + x)x - 0.19518(x \times x)x$ |
| Nguyen-3 | Truth | $y = x^5 + x^4 + x^3 + x^2 + x$ |
| | GSR | $0.5\,y = +0.125x + 0.25(x \times x \times x \times x) + 0.125(x)x + 0.5(x \times x)x$ $+0.125x + 0.5x(x \times x \times x \times x) + 0.125x + 0.125(x \times x) + 0.125x$ $+0.125(x)x + 0.125(x \times x) + 0.25x(x \times x \times x)$ |
| Nguyen-4 | Truth | $y = x^6 + x^5 + x^4 + x^3 + x^2 + x$ |
| | GSR | $-0.48318\,y = -0.096636x - 0.48318x(x \times x)(x \times x) - 0.16106(x \times x) - 0.16106(x \times x)$ $-0.24159(x \times x)x - 0.096636x - 0.48318(x \times x \times x)(x \times x)x$ $-0.24159(x \times x \times x) - 0.24159(x \times x \times x)(x + x) - 0.096636x$ $-0.096636x - 0.096636x - 0.16106(x \times x)$ |
| Nguyen-5 | Truth | $y = \sin(x^2)\cos(x) - 1$ |
| | GSR | $0.63246\,y = 0.63246\cos(x)\sin(x \times x) - 0.31623 - 0.31623$ |
| Nguyen-6 | Truth | $y = \sin(x) + \sin(x + x^2)$ |
| | GSR | $0.5\,y = 0.5\cos(x)\sin(x \times x) + 0.5\sin(x) + 0.5\cos(x \times x)\sin(x)$ |
| Nguyen-7 | Truth | $y = \ln(x + 1) + \ln(x^2 + 1)$ |
| | GSR | $0.70956\,e^y = +0.1095x(x \times x \times x) + 0.1095(x \times x \times x \times x) + 0.17082x + 0.23652$ $+0.17082x + 0.12702(x \times x)(x + x) + 0.17082x - 0.1095x(x + x)(x \times x)$ $+0.22776x(x \times x) + 0.23652 + 0.22776(x \times x)x + 0.23652$ $+0.23652(x + x + x)x + 0.17082x + 0.01314(x + x)$ |
| Nguyen-8 | Truth | $y = \sqrt{x}$ |
| | GSR | $0.83654\,\ln(y) = -0.032175\ln(x \times x \times x \times x) + 0.54697\ln(x)$ |
| Nguyen-9 | Truth | $y = \sin(x_1) + \sin(x_2^2)$ |
| | GSR | $-0.57735\,y = -0.57735\sin(x_1) - 0.57735\sin(x_2 \times x_2)$ |
| Nguyen-10 | Truth | $y = 2\sin(x_1)\cos(x_2)$ |
| | GSR | $0.44721\,y = 0.89442\sin(x_1)\cos(x_2)$ |
| Nguyen-11 | Truth | $y = x_1^{x_2}$ |
| | GSR | $0.70711\,\ln(y) = 0.70711\,x_2\ln(x_1)$ |
| Nguyen-12 | Truth | $y = x_1^4 - x_1^3 + \frac{1}{2}x_2^2 - x_2$ |
| | GSR | $-0.4\,y = -0.2(x_2 \times x_2) - 0.4x_1 - 0.4x_1 + 0.4(x_1 + x_2 + x_1) + 0.4(x_1 \times x_1)x_1$ $-0.4x_1(x_1 \times x_1 \times x_1)$ |
| Nguyen-12$^\star$ | Truth | $y = x_1^4 - x_1^3 + \frac{1}{2}x_2^2 - x_2$ |
| | GSR | $-0.6\,y = -0.1(x_2 + x_2 + x_2)x_2 - 0.3(x_1 + x_1)(x_1 \times x_1 \times x_1) + 0.3(x_1 \times x_1 \times x_1)$ $+0.3x_1(x_1 \times x_1) + 0.6x_2$ |

Table 22: Typical expressions (with exact symbolic equivalence) recovered by GSR for the Jin benchmark set. Note that the coefficients in the GSR expressions form a unit vector due to the normalization constraint imposed by the Lasso problem in Equation 6. It is easy to verify (by simplification) that the GSR expressions are symbolically equivalent to the ground truth expressions. Although GSR does not exactly recover Jin-6, it recovers approximations with very low RMSE, as shown in Table 5.

| Benchmark | | Expression |
|---|---|---|
| Jin-1 | Truth | $y = 2.5x_1^4 - 1.3x_1^3 + 0.5x_2^2 - 1.7x_2$ |
| | GSR | $0.30664\,y = +0.15332x_2^2 - 0.398632x_1^3 - 0.260644x_2 - 0.260644x_2 + 0.7666x_1x_1^3$ |
| Jin-2 | Truth | $y = 8x_1^2 + 8x_2^3 - 15$ |
| | GSR | $0.06909\,y = -0.518175 + 0.55272x_2^3 - 0.518175 + 0.27636x_1^2 + 0.27636x_1^2$ |
| Jin-3 | Truth | $y = 0.2x_1^3 + 0.5x_2^3 - 1.2x_2 - 0.5x_1$ |
| | GSR | $-0.7943\,y = -0.11914x_2 - 0.15886x_1^3 + 0.39715(x_2 + x_2 + x_2 + x_1) - 0.11915x_2 - 0.39715x_2^3$ |
| Jin-4 | Truth | $y = 1.5e^{x_1} + 5\cos(x_2)$ |
| | GSR | $-0.25198\,y = -0.37797e^{x_1} - 0.62995\cos(x_2) - 0.62995\cos(x_2)$ |
| Jin-5 | Truth | $y = 6\sin(x_1)\cos(x_2)$ |
| | GSR | $0.13484\,y = -0.80904\sin(x_2)\cos(x_1) + 0.40452\sin(x_2 + x_1) + 0.40452\sin(x_2 + x_1)$ |
| Jin-6 | Truth | $y = 1.35x_1x_2 + 5.5\sin((x_1 - 1)(x_2 - 1))$ |
| | GSR | Not exactly recovered |

Table 23: Typical expressions (with exact symbolic equivalence) recovered by GSR for the Neat benchmark set. Note that the coefficients in the GSR expressions form a unit vector due to the normalization constraint imposed by the Lasso problem in Equation 6. It is easy to verify (by simplification) that the GSR expressions are symbolically equivalent to the ground truth expressions. Although GSR does not exactly recover Neat-6, Neat-7, Neat-8, and Neat-9, it recovers approximations with very low RMSE, as shown in Table 6.

| Benchmark | | Expression |
|---|---|---|
| Neat-1 | Truth | $y = x^4 + x^3 + x^2 + x$ |
| | GSR | $-0.64952\,y = -0.32476x(x + x)(x \times x) - 0.064952(x + x) + 0.082996(x + x + x)$ $-0.32476(x \times x) - 0.2129x - 0.32476(x \times x) - 0.18764x(x + x)x$ $-0.064952(x + x) - 0.2129x - 0.2129x - 0.27424(x \times x \times x)$ |
| Neat-2 | Truth | $y = x^5 + x^4 + x^3 + x^2 + x$ |
| | GSR | $0.3914\,y = +0.3914x(x \times x \times x \times x) + 0.18274x + 0.27676x(x) + 0.18274x$ $+0.02794(x + x) + 0.3914x(x \times x) - 0.02702(x + x + x)(x + x)$ $+0.18274x + 0.27676(x \times x) - 0.12682(x + x + x) + 0.3914(x \times x \times x)x$ $+0.18274x + 0.18274x - 0.12682(x + x + x) + 0.18274x$ |
| Neat-3 | Truth | $y = \sin(x^2)\cos(x) - 1$ |
| | GSR | $-0.57735\,y = -0.57735\sin(x \times x)\cos(x) + 0.57735$ |
| Neat-4 | Truth | $y = \ln(x + 1) + \ln(x^2 + 1)$ |
| | GSR | $-0.5\,e^y = -0.25(x \times x) - 0.5 - 0.25x - 0.25x - 0.5x(x \times x) - 0.25(x \times x)$ |
| Neat-5 | Truth | $y = 2\sin(x_1)\cos(x_2)$ |
| | GSR | $0.57735\,y = 0.57735\cos(x_2)\sin(x_1) + 0.57735\cos(x_2)\sin(x_1)$ |
| Neat-6 | Truth | $y = \sum_{k=1}^{x} \frac{1}{k}$ |
| | GSR | Not exactly recovered |
| Neat-7 | Truth | $y = 2 - 2.1\cos(9.8x_1)\sin(1.3x_2)$ |
| | GSR | Not exactly recovered |
| Neat-8 | Truth | $y = \frac{e^{-(x_1-1)^2}}{1.2 + (x_2 - 2.5)^2}$ |
| | GSR | Not exactly recovered |
| Neat-9 | Truth | $y = \frac{1}{1 + x_1^{-4}} + \frac{1}{1 + x_2^{-4}}$ |
| | GSR | Not exactly recovered |

Table 24: Typical expressions (with exact symbolic equivalence) recovered by GSR for the Livermore benchmark set. Note that the coefficients in the GSR expressions form a unit vector due to the normalization constraint imposed by the Lasso problem in Equation 6. It is easy to verify (by simplification) that the GSR expressions are symbolically equivalent to the ground truth expressions. Although GSR does not exactly recover Livermore-7 and Livermore-8, it naturally recovers their Taylor approximations, as discussed in Appendix A.

| Benchmark | | Expression |
|---|---|---|
| Livermore-1 | Truth | $y = \frac{1}{3} + x + \sin(x^2)$ |
| | GSR | $0.65079\,y = 0.21693 + 0.65079\sin(x \times x) + 0.325395(x + x)$ |
| Livermore-2 | Truth | $y = \sin(x^2)\cos(x) - 2$ |
| | GSR | $-0.40825\,y = -0.40825\cos(x)\sin(x \times x) + 0.8165$ |
| Livermore-3 | Truth | $y = \sin(x^3)\cos(x^2) - 1$ |
| | GSR | $-0.57735\,y = 0.57735 - 0.57735\sin(x \times x \times x)\cos(x \times x)$ |
| Livermore-4 | Truth | $y = \ln(x+1) + \ln(x^2+1) + \ln(x)$ |
| | GSR | $-0.50998\,e^y = -0.25499(x \times x) - 0.21064x - 0.022175(x + x) - 0.21064x - 0.25499(x \times x)$ $-0.50998(x \times x)(x \times x) - 0.50998x(x \times x) - 0.022175(x + x)$ |
| Livermore-5 | Truth | $y = x_1^4 - x_1^3 + x_1^2 - x_2$ |
| | GSR | $0.44721\,y = -0.44721x_2 + 0.44721(x_1 \times x_1) - 0.44721(x_1 \times x_1)x_1 + 0.44721(x_1 \times x_1)(x_1 \times x_1)$ |
| Livermore-6 | Truth | $y = 4x^4 + 3x^3 + 2x^2 + x$ |
| | GSR | $-0.15763\,y = -0.26677x - 0.26677x - 0.093216(x + x) - 0.23918(x \times x)$ $-0.03804(x + x)x - 0.63052x(x)(x \times x) - 0.47289(x \times x \times x)$ $-0.093216(x + x) + 0.253886(x + x + x + x) - 0.26677x$ |
| Livermore-7 | Truth | $y = \sinh(x)$ |
| | GSR | Not exactly recovered |
| Livermore-8 | Truth | $y = \cosh(x)$ |
| | GSR | Not exactly recovered |
| Livermore-9 | Truth | $y = x^9 + x^8 + x^7 + x^6 + x^5 + x^4 + x^3 + x^2 + x$ |
| | GSR | $-0.30767\,y = -0.30767x(x \times x \times x)(x \times x) - 0.30767(x \times x \times x)x - 0.266671(x \times x)$ $-0.30767(x \times x \times x)(x)(x \times x \times x) - 0.153835 - 0.30767(x \times x \times x)$ $-0.30767(x \times x \times x \times x \times x)(x \times x \times x \times x) - 0.30767(x \times x \times x \times x)x$ $+0.056037(x + x + x + x)(x + x + x + x) - 0.266671(x \times x) - 0.153835x$ $-0.30767(x \times x \times x \times x)(x)(x \times x \times x) - 0.22364x(x + x + x)$ |
| Livermore-10 | Truth | $y = 6\sin(x_1)\cos(x_2)$ |
| | GSR | $0.22942\,y = 0.68826\cos(x_2)\sin(x_1) + 0.68826\sin(x_1)\cos(x_2)$ |
| Livermore-11 | Truth | $y = \frac{x_1^2 x_1^2}{x_1 + x_2}$ |
| | GSR | $-0.40825\,\ln(y) = -0.8165\ln(x_1 \times x_1) + 0.40825\ln(x_2 + x_1)$ |

Table 25: Typical expressions (with exact symbolic equivalence) recovered by GSR for the Livermore benchmark set (cont'd). Note that the coefficients in the GSR expressions form a unit vector due to the normalization constraint imposed by the Lasso problem in Equation 6. It is easy to verify (by simplification) that the GSR expressions are symbolically equivalent to the ground truth expressions.

| Benchmark | | Expression |
|---|---|---|
| Livermore-12 | Truth | $y = \frac{x_1^5}{x_2^4}$ |
| | GSR | $0.16903 \ln(y) = 0.84515 \ln(x_1) - 0.50709 \ln(x_2)$ |
| Livermore-13 | Truth | $y = x^{\frac{1}{3}}$ |
| | GSR | $0.97332 \ln(y) = 0.16222 \ln(x) + 0.16222 \ln(x)$ |
| Livermore-14 | Truth | $y = x^3 + x^2 + x + \sin(x) + \sin(x^2)$ |
| | GSR | $0.40825\, y = 0.40825x(x \times x) + 0.40825\sin(x) + 0.40825(x \times x) + 0.40825\sin(x \times x) + 0.40825x$ |
| Livermore-15 | Truth | $y = x^{\frac{1}{5}}$ |
| | GSR | $0.99015 \ln(y) = 0.099015 \ln(x) + 0.099015 \ln(x)$ |
| Livermore-16 | Truth | $y = x^{\frac{2}{5}}$ |
| | GSR | $0.99504 \ln(y) = 0.099504 \ln(x \times x \times x \times x \times x)$ |
| Livermore-17 | Truth | $y = 4\sin(x_1)\cos(x_2)$ |
| | GSR | $0.17408\, y = 0.69632 \sin(x_2 + x_1) - 0.69632\cos(x_1)\sin(x_2)$ |
| Livermore-18 | Truth | $y = \sin(x^2)\cos(x) - 5$ |
| | GSR | $-0.19245\, y = 0.96225 - 0.19245\cos(x)\sin(x \times x)$ |
| Livermore-19 | Truth | $y = x^5 + x^4 + x^2 + x$ |
| | GSR | $-0.46202\, y = -0.46202x(x \times x)x - 0.015609(x + x + x) + -0.46202 * (x^1 * x^1) - 0.46202x(x \times x)(x \times x)$ $-0.290313x - 0.15297(x + x) - 0.15297(x + x) + 0.12175(x + x + x + x)$ |
| Livermore-20 | Truth | $y = e^{-x^2}$ |
| | GSR | $-0.89442 \ln(y) = 0.44721(x + x)x$ |
| Livermore-21 | Truth | $y = x^8 + x^7 + x^6 + x^5 + x^4 + x^3 + x^2 + x$ |
| | GSR | $-0.38914\, y = -0.027357(x + x + x)x - 0.38914(x \times x \times x)(x \times x) - 0.38914x(x \times x \times x \times x \times x)(x \times x \times x)$ $+0.0714425x(x + x)(x + x) - 0.38914x - 0.38914(x \times x \times x \times x \times x) - 0.22497(x \times x \times x)$ $-0.19457(x \times x \times x \times x \times x)(x + x)(x \times x) - 0.22497x(x \times x) - 0.027357(x + x + x)$ $-0.22497(x \times x)x - 0.224998(x \times x) - 0.097285x(x + x + x + x)(x \times x \times x \times x)$ |
| Livermore-22 | Truth | $y = e^{-0.5x^2}$ |
| | GSR | $0.8165 \ln(y) = -0.40825(x + x)x + 0.40825(x \times x)$ |

Table 26: Typical expressions (with exact symbolic equivalence) recovered by GSR for the SymSet benchmark set. Note that the coefficients in the GSR expressions form a unit vector due to the normalization constraint imposed by the Lasso problem in Equation 6. It is easy to verify (by simplification) that the GSR expressions are symbolically equivalent to the ground truth expressions.

| Benchmark | | Expression |
|---|---|---|
| SymSet-1 | Truth | $y = x\sinh(x) - \frac{4}{5}$ |
| | GSR | $0.70448\,y = 0.35224e^x x - 0.17612xe^{-x} - 0.563584e^x e^{-x} - 0.17612xe^{-x}$ |
| SymSet-2 | Truth | $y = (x^5 - 3x^4 - 2.8x + 5)^{-1}$ |
| | GSR | $-0.11335\,y^{-1} = -0.23188x - 0.11335 + 0.50651(x+x) - 0.11335 - 0.23188x$ $-0.11335 - 0.11335 - 0.23188x - 0.11335(x \times x \times x \times x)x$ $-0.11335 + 0.34005x(x \times x \times x)$ |
| SymSet-3 | Truth | $y = (x^4 - 1.2x^2 + 11.5)^{\frac{1}{3}}$ |
| | GSR | $0.1173\,y^3 = +0.26576x^2 + 0.26576x^2 + 0.26576(x \times x) + 0.44965 + 0.44965$ $+0.02346(x+x+x+x+x)x^3 - 0.23451(x+x+x+x)x + 0.44965$ |
| SymSet-4 | Truth | $y = 0.8 - \cos(x) + 4.2e^x\sin(x^2)$ |
| | GSR | $0.22485\,y = -0.112425\cos(x) + 0.94437\sin(x \times x)e^x - 0.112425\cos(x) + 0.17988$ |
| SymSet-5 | Truth | $y = 4.5x_1^2 + x_1 x_2^3 - 1.7x_2 - 3.1$ |
| | GSR | $-0.16964\,y = -0.16964(x_2 \times x_2 \times x_2 \times x_1) - 0.76338(x_1 \times x_1) + 0.525884 + 0.288388x_2$ |
| SymSet-6 | Truth | $y = \frac{5}{3x_1 - x_2^3}$ |
| | GSR | $0.84515\,y^{-1} = -0.16903(x_2 \times x_2)x_2 + 0.50709x_1$ |
| SymSet-7 | Truth | $y = \ln(x_1^3 + 4x_1 x_2)$ |
| | GSR | $-0.482715\,e^y = -0.64362(x_1 + x_1 + x_1)x_2 + 0.21883x_2 - 0.482715(x_1 \times x_1 \times x_1) - 0.21883x_2$ |
| SymSet-8 | Truth | $y = \sqrt{5x_1^5 + 14x_1^3 x_2^4 - 2x_2 + 7}$ |
| | GSR | $0.060634\,y^2 = 0.30317(x_1^2 \times x_1^3) + 0.848876x_1^2 x_2^3(x_1 \times x_2) + 0.424438 - 0.060634(x_2 + x_2)$ |
| SymSet-9 | Truth | $y = (2x_1 + x_2)^{-\frac{2}{3}}$ |
| | GSR | $0.83205\,\ln(y) = -0.5547\ln(x_1 + x_2 + x_1)$ |
| SymSet-10 | Truth | $y = 1.5\cos(x_1)\ln(x_1 x_2) - 2.5$ |
| | GSR | $0.32444\,y = -0.8111 + 0.48666\ln(x_1 \times x_2)\cos(x_1)$ |
| SymSet-11 | Truth | $y = \sqrt{2\cos(x_1) + 30e^{x_2}} + 4$ |
| | GSR | $-0.228568\,y + 0.028571\,y^2 = -0.457136 + 0.85713e^{x_2} + 0.057142\cos(x_1)$ |
| SymSet-12 | Truth | $y = 0.4x_1^4 + 6.2x_2 - 3.5x_1 x_3 - 4.5$ |
| | GSR | $0.16288\,y = -0.18324 + 0.065152x_1(x_1)(x_1 \times x_1) - 0.57008(x_3 \times x_1) + 0.504928x_2 - 0.18324$ $-0.18324 - 0.18324 + 0.504928x_2$ |
| SymSet-13 | Truth | $y = \frac{2x_2}{x_1 + x_3}$ |
| | GSR | $0.57735\,\ln(y) = 0.57735\ln(x_2 + x_2) - 0.57735\ln(x_1 + x_3)$ |
| SymSet-14 | Truth | $y = \frac{x_1 x_2 x_3}{x_1 + x_2 + x_3}$ |
| | GSR | $0.57735\,\ln(y) = 0.57735\ln(x_1 \times x_2 \times x_3) - 0.57735\ln(x_2 + x_3 + x_1)$ |
| SymSet-15 | Truth | $y = (x_1 + x_2)^{x_3}$ |
| | GSR | $0.70711\,\ln(y) = 0.70711\,x_3\ln(x_1 + x_2)$ |
| SymSet-16 | Truth | $y = e^{2.6x_1 - \ln(x_2) + 9.8\cos(x_3)}$ |
| | GSR | $0.098035\,\ln(y) = 0.960743\cos(x_3) - 0.0490175\ln(x_2 \times x_2) + 0.254891x_1$ |
| SymSet-17 | Truth | $y = \ln\left(0.2e^{x_1 + x_2} + 0.5\cos(x_3^2)\right)$ |
| | GSR | $0.88045\,e^y = 0.17609e^{x_1 + x_2} + 0.440225\cos(x_3 \times x_3)$ |

## E    Limitations

Although GSR achieves great results whether by recovering exact expressions or approximations with low errors, it still has several limitations:

**Absence of division operations.** The primary limiting factor to our GSR method is that it still cannot handle divisions. This is due to the way we define our encoding scheme. In this current version, we only consider a weighted sum of basis functions where the basis functions are a product of transformations; no divisions are involved. We can overcome this issue by modifying the encoding scheme to include divisions within the basis functions $\big($e.g. a negative integer in the first column of the basis functions implies a division by the corresponding transformation, i.e. using Table 13, a first-column entry of $-8$ encodes the division $\frac{1}{\ln}\big)$. However, this will significantly increase the total number of possible combinations in which we can form basis matrices. Due to the lack of divisions in its current version, GSR suffers on some benchmarks such as Neat-6, Neat-8, Neat-9, Livermore-7, Livermore-8. It is worth noting that GSR can recover some divisions with the help of the ln or $\bullet^{-1}$ operators (see Livermore-11, Livermore-12, Livermore-20, Livermore-22, SymSet-2, SymSet-6, SymSet-9, SymSet-13, and SymSet-14). This is only possible when the original function consists of only one term (not a sum of terms).

**Composition of tranformations.** Another limiting factor to the current version of GSR is that it cannot recover expressions containing composite functions, such as $y = e^{\cos(x)} + \ln(x)$. In this example, the basis function $e^{\cos(x)}$ cannot be recovered by GSR due to our encoding scheme. Again, if $\ln(x)$ was not there, that is, if the function contained the first term only, i.e. $y = e^{\cos(x)}$, then GSR can handle the situation by recovering $\ln(y) = \cos(x)$. The benefits of using $g(y) = f(\mathbf{x})$ can be clearly observed on the SymSet benchmark problems (especially SymSet-16, and SymSet-17).

**Choice of hyperparameters and search process.** Throughout our experiments, we have observed that, for some benchmarks (such as Jin-6 and Neat-7), althought they are expressible by GSR, they were not fully recovered. GSR only recovered approximations for these benchmarks with very low errors. This can be explained by two reasons: i) The choice of hyperparameters affects the search process, ii) Our matrix-based GP search process may not be very effective on these benchmarks, given the complexity of their corresponding basis functions, and thus they may require a huge number of iterations to be recovered. That is, if we keep our code running for a very long time, we may be able to recover these benchmarks. This can be verified by expanding Jin-6 and Neat-7 as follows:

Jin-6:
$$
\begin{aligned}
y &= 1.35x_1x_2 + 5.5\sin\left((x_1 - 1)(x_2 - 1)\right)\\
&= 1.35x_1x_2 + 5.5\sin\left(x_1x_2 - x_1 - x_2 + 1\right)\\
&= 1.35x_1x_2 + 5.5\big[\sin\left(-x_1 - x_2\right)\cos\left(x_1x_2 + 1\right) + \cos\left(-x_1 - x_2\right)\sin\left(x_1x_2 + 1\right)\big]\\
&= 1.35x_1x_2 - 5.5\cos(1)\underbrace{\sin(x_1 + x_2)\cos(x_1x_2)}_{\phi_1(\mathbf{x})} + 5.5\sin(1)\underbrace{\sin(x_1 + x_2)\sin(x_1x_2)}_{\phi_2(\mathbf{x})}\\
&\quad + 5.5\cos(1)\underbrace{\cos(x_1 + x_2)\sin(x_1x_2)}_{\phi_3(\mathbf{x})} + 5.5\sin(1)\underbrace{\cos(x_1 + x_2)\cos(x_1x_2)}_{\phi_4(\mathbf{x})}
\end{aligned}
$$

Neat-7:
$$
\begin{aligned}
y &= 2 - 2.1\cos(9.8x_1)\sin(1.3x_2)\\
&= 2 - 2.1\big(\cos(9.8)\cos(x_1) - \sin(9.8)\sin(x_1)\big)\big(\sin(1.3)\cos(x_2) + \cos(1.3)\sin(x_2)\big)\\
&= 2 - 2.1\cos(9.8)\sin(1.3)\underbrace{\cos(x_1)\cos(x_2)}_{\phi_1(\mathbf{x})} + 2.1\sin(9.8)\sin(1.3)\underbrace{\sin(x_1)\cos(x_2)}_{\phi_2(\mathbf{x})}\\
&\quad - 2.1\cos(9.8)\cos(1.3)\underbrace{\cos(x_1)\sin(x_2)}_{\phi_3(\mathbf{x})} + 2.1\sin(9.8)\cos(1.3)\underbrace{\sin(x_1)\sin(x_2)}_{\phi_4(\mathbf{x})}
\end{aligned}
$$

As we can see, GSR has to find the four corresponding basis functions simultaneously in order to recover the expressions.

Indeed, there are plenty of expressions that still cannot be fully recovered by our GSR method. This is the case for all the other methods as well.

