# OpenReview forum: "GSR: A Generalized Symbolic Regression Approach"
_TMLR — Accepted by TMLR_

### Review · Reviewer_QcjW · 2022-10-28

**Summary Of Contributions:**

This paper develops a new approach for symbolic regression. The approach combines advantages from the usually disparate fields of sparse system identification (SINDy) and genetic programming (GP). This involves a new matrix representation for basis functions. GP is used to generate sets of such basis functions which are then plugged into a classical algorithm for finding sparse linear combinations of basis functions. The paper also introduces a reformulation of the symbolic regression (SR) objective that makes search smoother under their method, and makes their search space more expressive. The paper introduces a new benchmark that highlights the effect of this reformulation. In detailed experiments, the new method performs well compared to several key alternative SR methods on popular benchmarks.

**Audience:**

Yes

**Claims And Evidence:**

Yes

**Requested Changes:**

Changes critical to securing my recommendation for acceptance:
- Explicit mention of connections to related LASSO-like methods (e.g., SINDy methods) and finding invariants with Eureqa.

Changes not critical to securing my recommendation for acceptance, but that would strengthen the work:
- Move the "Making Predictions" section of the Discussion to the methods section, since the reader wonders this immediately. I'd put it at the end of Section 3.1
- SymSet should be described at least at a high-level in the main paper, i.e., what are it's key properties that other benchmarks don't have? and what was the process for coming up with the problems in the benchmark?
- It would be great to see some plot of how the evolutionary algorithm converges for at least some problems (at least in the appendix). This would give a much clearer sense of what the complete method is actually doing. Ideally, you could even show details of how the set of basis functions is refined over iterations.
- The reader immediately is interested in seeing results for the alternate methods (NGGPPS, DSR, ...) on SymSet.

**Strengths And Weaknesses:**

Strengths:

- The paper is well-motivated, and includes solid background and related work.
- The experiments are solid and thorough: good comparison methods and benchmarks used.
- The proposed method provides a first answer to an important open question: How to combine evolutionary methods with more classical system identification (sparse non-linear regression) methods.
- Even though each of the algorithm components are relatively simple, the results are quite strong, highlighting many possibilities for future improvement.

Weaknesses:

- Although sparse system identification methods are cited, their technical connection is not mentioned in the methods section. These methods also use some LASSO-like methods for finding solutions that are linear combinations of basis functions. As is, it is not clear from the paper that such techniques are already well-developed.
- Similarly, the reformulated version of SR is related to the key method in Eureqa of discovering invariants instead of trying to do prediction directly. Do the authors agree that this connection is related enough to be worth mentioning?
- The new SymSet benchmark is useful to show importance of the SR reformulation, but it's not clear from the main paper how this benchmark was designed, e.g., what properties it is intended to have that other benchmarks do not. The reader would also like to see evaluation of the comparison methods on this new benchmark.
- There are no plots of the GP convergence process, so it is difficult for the reader to get a feel for how the method behaves.
- Although the proposed method is very practical compared to other SR methods today, it still avoids the key challenge in SR: how to find deep compositions of functions. The paper acknowledges this limitation, and says it will be addressed in future work, but does not suggest how it could be overcome, or why this new approach is a particularly good platform for handling this.

---

> ### Author Response · Authors · 2022-12-01
> **Response to Reviewer QcjW**
>
> We thank the reviewer for recognizing the novelty and contribution presented in the paper, and for providing valuable comments to improve the exposition of the paper!\
> Please check our revised paper.
>
> As requested by the reviewer, we have made the following changes:
>
> $\bullet$ We fully agree with the reviewer about the connections between our proposed method and related LASSO-like methods such as SINDy methods, as well as finding invariants with Eureqa. To reflect these connections, we have explicitly mentioned them in Section 3, and we have added two paragraphs in the Discussion section (“Closely related work”, and “GSR: a simple yet promising algorithm”).\
> $\bullet$ We have moved the “Making Predictions” section of the Discussion to the end of Section 3.1.\
> $\bullet$ Note that SymSet contains benchmark problems with similar properties as Nguyen, Jin, Neat, and Livermore benchmarks, but with an additional function composition (or symbolic layer), which can explicitly highlight the benefits we gain from using our proposed GSR approach; by simply applying inverse transformations on the output, the problem reduces to a single symbolic layer task. We have made this clear in the first paragraph of Section 4, thank you for pointing this out!\
> $\bullet$ Regarding GSR’s limitation on deep composition of functions, we have proposed some ways to overcome it in the “Limitations” paragraph of the Discussion section. In short, one could modify the search space either by expanding the definition of a basis function to account for composition of functions up to some number of layers, or by completely modifying the search space to a symbolic neural network as in [1,2,3].\
> $\bullet$ To get a sense of how our GSR method compares to other strong SR methods, we have conducted additional experiments on SymSet. We have applied many strong SR benchmark methods: NGGPPS [4] (which is an improved and more generalized version of DSR [6]), GPLearn (Koza-style [6] SR method in Python, staged for benchmarking), PySR [7,8], and PSTree [9] (which achieves state-of-the-art performance). We reported the results in Table 8 of the revised paper (with full details reported in the appendix).
>
> $\textbf{References}$\
> [1] Extrapolation and learning equations. 2016. arXiv:1610.02995.\
> [2] Learning Equations for Extrapolation and Control. ICML 2018.\
> [3] Integration of Neural Network-Based Symbolic Regression in Deep Learning for Scientific Discovery. IEEE Transactions on Neural Networks and Learning Systems. 2021.\
> [4] Symbolic Regression via Neural-Guided Genetic Programming Population Seeding. NeurIPS 2021.\
> [5] Deep Symbolic Regression: Recovering Mathematical Expressions from Data via Risk-Seeking Policy Gradients. ICLR 2021.\
> [6] J. Koza. Genetic Programming: On the Programming of Computers by Means of Natural Selection. 1992.\
> [7] M. Cranmer. PySR: Fast & Parallelized Symbolic Regression in Python/Julia. Zenodo. 2020.\
> [8] Discovering Symbolic Models from Deep Learning with Inductive Biases. NeurIPS 2020.\
> [9] PS-Tree: A Piecewise Symbolic Regression Tree. Swarm and Evolutionary Computation. 2022.

---

> > ### Comment · Reviewer_QcjW · 2022-12-13
> > **Thanks for the response**
> >
> > Thanks for the response and updates. The paper now is technically well-substantiated.
> >
> > One more overall note for the authors (it won't affect my recommendation for acceptance): The main remaining place where the story gets difficult for the reader is that the paper is framed around the importance of the generalization of SR, but the paper not only introduces this generalization, it also introduces a new SR algorithm, i.e., new combination of representation, evolution, and coefficient optimization. The reader wonders what would happen if the generalized SR approach were simply applied to an existing SOTA SR method like Operon. In other words, is the generalized approach generally useful for SR, or just for this particular algorithm? At this point, this question will be left for future work.

---

> > > ### Author Response · Authors · 2022-12-21
> > > **Applying GSR's concept to existing SR methods**
> > >
> > > Thank you for your response! Please note that extending GSR's concept to existing SR methods is definitely feasible.
> > >
> > > In fact, this was explicitly mentioned at the end of the Conclusion section as follows:\
> > > "We believe that GSR’s concept of fitting $g(y) = f(x)$ can be applied to other SR methods and can boost their performance."
> > >
> > > That said, this may require some modifications to the parameter/coefficient optimization process; this could be an interesting research direction for future work.

---

### Review · Reviewer_ZWfR · 2022-11-04

**Summary Of Contributions:**

This work proposes a generalized symbolic regression (GSR) approach to learn the mathematical expression from a set of independent variables to a target variable for a given dataset. The traditional symbolic regression (SR) approach usually has the form y = f(x) where f is a learnable analytical function. The proposed GSR is to learn the form g(y) = f(x) where g is also a learnable analytical function.

For the actual algorithm, this work formulates both g and f as a linear weighted sum of different learnable basis functions with a matrix-based encoding scheme. A genetic programming method and an ADMM-based optimization method are applied to 1) find the set of basis functions and 2) determine the weights for each basis function, respectively. Experimental results show GSR can achieve promising performance on some SR benchmarks and a newly proposed challenging SymSet dataset.


**Audience:**

Yes

**Claims And Evidence:**

Yes

**Requested Changes:**

Please address the major concerns listed in the weaknesses. It is crucial to:

1) Analyze and discuss GSR's expression ability, and show why it could be better than the SR counterpart with similar expression ability such as y = h(f(x))

2) Conduct more solid and convincing experiments to validate the performance of GSR with strong SR baselines on more realistic problems.


**Strengths And Weaknesses:**

**Strengths**

- This work is generally well-organized and easy to follow.
- Symbolic regression is an important and challenging problem in machine learning. Some TMLR audiences could be interested in this work.
- The proposed GSR approach is straightforward and has achieved promising experimental performance.

**Weaknesses**

The TMLR guideline asks the reviewer to check the following:

(a) Are the claims made in the submission supported by accurate, convincing and clear evidence?

(b) Would some individuals in TMLR's audience be interested in the findings of this paper?

Based on the strengths of this paper, I believe the requirement (b) is well satisfied, but some claims made in this submission are not well supported.


**1. The Claimed Generalized SR**

This work claims the proposed GSR approach generalizes the traditional SR approach, which searches for the form g(y) = f(x) instead of y = f(x). However, the proposed g(y) = f(x) formulation is very similar to the simple composition function formulation such as y = h(f(x)). If g() is invertible, we could have h = g^-1. Let f_new = hf, y = f_new(x) is the standard SR. What is the advantage of the proposed g(y) = f(x) formulation over y = h(f(x))?

Even when g() is not invertible, it seems that it is still straightforward to find a function class h() such that y = h(f(x)) has a similar expression ability with g(y) = f(x). As correctly pointed out in this work, when g() is not invertible, the g(y) = f(x) formulation could indeed lead to an undesirable inference approach (e.g., a root-finding algorithm is needed).

Without a thorough analysis and discussion, it is hard to evaluate the actual advantage and contribution of the proposed GSR approach.

**2. Linear Combination and Expression Ability**

This work restricts the f(x) and g(y) as a linear weighted sum combination of basis functions, but does not discuss its expression ability (with other methods). For example, given the same basic arithmetic operations and mathematical function, the formulation g(y) = f(x) should have better expression ability than y = f(x) (e.g., the s-GSR approach in the appendix). Therefore, it is expected that GSR should have better performance, especially when the ground truth function can be exactly represented by g(y) = f(x) but not y = f(x).

However, the crucial question would be the comparison with a more fair SR counterpart with similar expression ability. It is always possible to improve the expression ability of y = f(x), such as (1) using the composition formulation y = h(f(x)), (2) enriching the mathematical function, and (3) relaxing the linear combination constraint. Will GSR still outperform those SR counterparts? Does the current better performance of GSR actually come from its new formulation that cannot be achieved by the SR approach with similar expression ability?

The expression ability for other SR methods are not analyzed or discussed in this work.

**3. Experiment Setting and Comparison with Other Methods**

This work claims the current SR benchmarks (e.g., Nguyen, Jin, and Neat) are too easy for GSR and cannot really reflect its strengths. In this case, why not validate GSR's performance on the more realistic and strong SRBench benchmarks[1]?

GSR is compared with four SR methods, namely, NGGPPS, DSR, BSR, and Neat-GP. However, according to [1], the other SR methods such as Operon [2], AIFeynman [3], and AFP_FE [4] could have much better performances for different problems. Is there any reason to choose NGGPPS/DSR/BSR/Neat-GP as baselines, and not compare with Operon/AIFeynman/AFP_FE or other methods with strong performance in [1]?

**Other Comments**

- The proposed GSR has 17 different hyperparameters. Will its performance be sensitive to those hyperparameter settings?

- The function expression found by different SR methods will have different formula complexity, even when they are exact for the same ground truth expression. Will the GSR formulation lead to a higher formula complexity than other methods?

**Reference**

[1] Contemporary Symbolic Regression Methods and their Relative Performance. NeurIPS 2021.

[2] Parameter identification for symbolic regression using nonlinear least squares. Genetic Programming and Evolvable Machines 2019.

[3] AIFeynman 2.0: Pareto-optimal symbolic regression exploiting graph modularity. NeurIPS 2020.

[4] Distilling free-form natural laws from experimental data. Science 2009.

---

> ### Author Response · Authors · 2022-12-01
> **Response to Reviewer ZWfR**
>
> We thank the reviewer for their time and valuable comments.
>
> $\textbf{1. The Claimed Generalized SR}$\
> $\bullet$ The term “Generalized” in GSR mainly stands for its ability to discover analytical mappings from the input space to a transformed output space through expressions of the form g(y) = f(x). This generalizes the classical SR task of identifying expressions of the form y = f(x) (i.e. the latter is simply a special case of GSR with $g(y) = y$). In addition, the term “Generalized” can denote the fact that we constrain the search space to generalized linear models (as pointed out by Reviewer QcjW), keeping in mind that the search space could be confined to other generalized spaces.\
> $\bullet$ We agree that GSR could resemble a traditional SR task with formulation y = h(f(x)). However, we do not agree that this composition function task is “simple”, as claimed by the reviewer. GSR takes advantage of the fact that the target y is a scalar, and hence, we can avoid searching for functions h(.) in a space that could grow exponentially with the dimension of the input by simply searching for their corresponding inverse transformations in the output space. This concept, which happens implicitly, provides an edge for GSR over traditional SR methods in terms of runtime, complexity, and smoothness of search space (as Reviewer QcjW pointed out).
>
> $\textbf{2. Linear Combination and Expression Ability}$\
> $\bullet$ We agree that it is always possible to improve the expression ability of y = f(x), by:\
> (1) using composition formulation y = h(f(x)),\
> (2) enriching the mathematical function,\
> (3) relaxing the linear combination constraint.\
> However, all these suggested ways result in a complex search space that becomes intractable when the dimension of the input grows.
> In general, GSR attempts to transform an L-layer SR task into an (L-1)-layer SR task by simply transforming the output space. As currently presented in the paper, GSR is transforming a 2-layer SR task into a simple 1-layer task. That is, we overcome the burden of searching for composition of functions by searching for simple functions in a smoother and reduced space. We have added a paragraph in the Discussion section (“GSR's expression ability”) to address the concerns above.\
> $\bullet$ We agree that a comparison between GSR and a more fair SR counterpart with similar expression ability would improve the quality of the paper. To this end, we have run more experiments on SymSet using several strong SR benchmark methods (which even have better expression ability than GSR). In addition to GSR and s-GSR (see Table 8 in the paper), we have conducted experiments on NGGPS [1] (which improves on DSR [2]), GPLearn (a Koza-style [3] SR method), PySR [4], and PSTree [5] (which achieves sota performance). We reported the results in Table 8 of our revised paper (full details are in the appendix).
>
> $\textbf{3. Experiment Setting and Comparison with Other Methods}$\
> $\bullet$ Existing datasets are either too easy for GSR (e.g. Nguyen, Jin) or too complex (e.g. Feynman) to achieve satisfactory exact recovery rates. In both cases, these datasets do not really highlight the benefits of GSR. We proposed SymSet to fill the gap between those datasets while conveying the main message of this paper in a clear manner. SymSet contains benchmark problems with similar properties as Nguyen, Jin, Neat, and Livermore benchmarks, but with an additional function composition. We have explicitly mentioned this description at the beginning of Section 4.\
> $\bullet$ We agree that there are many SR methods that could have strong performances on different problems (e.g. Operon, AIFeynman, and AFP_FE). In our paper, we mainly benchmarked against NGGPPS [1] and DSR [2], following their experimental protocols and evaluation metrics (e.g. exact recovery rate, RMSE). The reviewer may have missed PSTree [5] which is the current top performer (beating Operon/AIFeynman/AFP_FE). As mentioned above, we have additionally compared GSR against PSTree and two other methods (i.e. PySR and gplearn). See results in Table 8.
>
> $\textbf{Other Comments}$\
> $\bullet$ As most of ML algorithms, GSR is sensitive to the hyperparameter settings. The numbers in Table 9 correspond to the tuned hyperparameters.\
> $\bullet$ Regarding the expression complexity, as we are limiting the maximum number of basis functions applied on the inputs and outputs, the GSR formulation lead to “reasonable” formula complexity compared to other methods (see tables 21-26).
>
> $\textbf{References}$\
> [1] Symbolic Regression via Neural-Guided Genetic Programming Population Seeding. NeurIPS 2021.\
> [2] Deep SR: Recovering Mathematical Expressions from Data via Risk-Seeking Policy Gradients. ICLR 2021.\
> [3] GP: On the Programming of Computers by Means of Natural Selection. 1992.\
> [4] PySR: Fast & Parallelized Symbolic Regression in Python/Julia. Zenodo. 2020.\
> [5] PS-Tree: A Piecewise Symbolic Regression Tree. Swarm and Evolutionary Computation. 2022.\

---

> > ### Comment · Reviewer_ZWfR · 2022-12-13
> > **Thank you for your thorough response**
> >
> > Thank you for your thorough response and the new discussions/experiments. All my concerns have been properly addressed and I lean toward accepting this paper.

---

### Review · Reviewer_zEUp · 2022-11-16

**Summary Of Contributions:**

The paper presents an approach called Generalized Symbolic Regression (GSR) trying to generalize the classical Symbolic Regression task. In particular the difference with respect to classical approaches is that the mapping is not directly done to the target variable but to a its transformation.


**Audience:**

Yes

**Broader Impact Concerns:**

No ethical implications

**Claims And Evidence:**

Yes

**Requested Changes:**

A more accurate experimental evaluation considering other methods and datasets could  improve the quality and the novelty of the paper.

The results are promising. As regards the time complexity it should be reported (sometimes the execution time could depend on the typical implementation and on the adopted programming language).

Another approach that can be used in the experimental evaluation should be that of reporting the complexity of the symbolic representation of the formula when compared to other methods.

Finally, instead of using the recovery rate, it should be interesting to consider the symbolic solution definition reported in [1] (def 4.1) designed to capture models that differ from the true model by a constant or scalar.

[1] Contemporary Symbolic Regression Methods and their Relative Performance

**Strengths And Weaknesses:**

The proposed approach is well described in the paper. Its formulation is quite simple and reflects other previous works using matrices and genetic algorithms.

The definition of the method is well reported in the paper by some definitions.

The final part of the paper regarding the experimental evaluation should be extended considering other sota algorithms for solving the same problem. See for instance the paper "Contemporary Symbolic Regression Methods and their Relative Performance" for other methods and datasets.

---

> ### Author Response · Authors · 2022-12-01
> **Response to Reviewer zEUp**
>
> We thank the reviewer for the positive feedback and helpful comments.
>
> $\bullet$ We totally agree with the reviewer that experimenting with other sota algorithms for solving the same problem would improve the quality and novelty of the paper. Accordingly, we have conducted an extensive experimental evaluation on the SymSet problem set using several strong SR benchmark methods. More specifically, in addition to GSR and s-GSR (see Table 8 in the paper), we have run experiments on NGGPS [1] (which improves on DSR [2]), GPLearn (which is a Koza-style [3] SR method in Python), PySR [4,5], and PSTree [6] (which achieves state-of-the-art performance). We reported the results in Table 8 of our revised paper (with full details reported in the appendix).\
> $\bullet$ Regarding time complexity, due to the inherently chaotic nature of most genetic algorithms, calculating/approximating time complexity with O(.) notation is unlikely to be useful (it could probably be misleading). Accordingly, for the purpose of this paper, we believe that an acceptable measure is the actual runtimes and their averages over multiple runs.\
> $\bullet$ Regarding the complexity of the symbolic representation, since each method has its own search space, and since some of the methods allow for composition of functions, measuring the expression complexity (by adopting some consistent measure) could be unfair. For instance, the more depth is included in a method (by means of function composition), the more likely the stopping criteria could be reached with lower expression complexity compared to methods with no function compositions. In our paper, we follow the same experimental protocols and evaluation criteria as two recent (and strong) SR methods  [1,2], i.e. we report recovery rates, RMSEs, as well as runtimes for a fair evaluation.\
> $\bullet$ Thank you for pointing out the symbolic solution definition reported in [7]. We also believe that it is an interesting metric and it would be worth adopting in our future work.
>
> $\textbf{References}$\
> [1] Symbolic Regression via Neural-Guided Genetic Programming Population Seeding. NeurIPS 2021.\
> [2] Deep Symbolic Regression: Recovering Mathematical Expressions from Data via Risk-Seeking Policy Gradients. ICLR 2021.\
> [3] J. Koza. Genetic Programming: On the Programming of Computers by Means of Natural Selection. 1992.\
> [4] M. Cranmer. PySR: Fast & Parallelized Symbolic Regression in Python/Julia. Zenodo. 2020.\
> [5] Discovering Symbolic Models from Deep Learning with Inductive Biases. NeurIPS 2020.\
> [6] PS-Tree: A Piecewise Symbolic Regression Tree. Swarm and Evolutionary Computation. 2022.\
> [7] Contemporary Symbolic Regression Methods and their Relative Performance. NeurIPS 2021.

---

### Decision · Action_Editors · 2022-12-17

**Recommendation:** Accept with minor revision

**Comment:**

Three knowledgeable reviewers agreed that the proposed GSR is novel and the claims in the paper (post rebuttal) are supported by evidence. Only some minor points needs to be addressed in the revision before publication:

  - **Disentangling contributions** I agree with reviewer QcjW in staying that, in principle, one could apply GSR in conjunction with other classical SR methods. I recommend authors to (briefly) discuss this aspect in the paper. Authors can also mention that "Generalized" here is to be intended as in "Generalized linear models" as also pointed out by reviewers ZWfR and QcjW.
  - **Add a discussion on computational complexity** I agree with reviewer zEUp that a discussion about computational complexity of GSR and competitors is missing. Especially one comparing GSR w.r.t. vanilla SR with the same genetic algorithm. I also agree with authors that big-O notation for worst-case complexity can be too pessimistic for genetic algorithms. However, it can be still valuable to understand how a method operates and scales.



**Audience:**

SR is a very relevant learning task that can be of interest for many researchers in ML and AI. Furthermore, methods employed in this paper can also be potentially interesting for the overlapping community of people researching symbolic search via optimization and genetic algorithms.

**Claims And Evidence:**

This paper tackles the important task of symbolic regression (SR), i.e., learning regression models explicitly represented as mathematical (symbolic) expressions. Specifically, authors propose a generalized symbolic regression (GSR) task, in which instead of learning the observed label response as a function of the input variables (i.e.,  y = f(x)), a transformation of the labels is also learned (i.e.,g(y) = f(x)).

Authors initially claimed that GSR was yielding the current state-of-the-art performance among SR methods. This is a very general claim. Experiments were run on a number of benchmarks for SR, showing the good performance of GSR w.r.t. strong baselines but omitting more recent and versatile SR competitors (e.g. Operon, AIFeynman, AFP_FE). After the rebuttal, the authors amended the initial claim of achieving state-of-the-art performance with a more fitting claim of showing promising competitive performance for SR on a number of sensible benchmarks.

The claims reported in the latest version of the manuscript are well supported by the current experimental setting, and GSR is a nice addition to the vast literature of SR.

---

> ### Author Response · Authors · 2022-12-30
> **Camera Ready Version Submitted**
>
> We would like to thank all reviewers and AE for handling our submission! We have submitted the camera-ready version.